Subject Areas:
psychology/behaviour/developmental biology

Keywords:
implicit theory of mind, replication, anticipatory looking, false belief

Author for correspondence:
Louisa Kulke
e-mail: lkulke@uni-goettingen.de

# Is implicit Theory of Mind real but hard to detect? Testing adults with different stimulus materials

Louisa Kulke[1,2,3], Marieke Wübker[1,4] and Hannes Rakoczy[1,3]

[1]Department of Developmental Psychology, University of Göttingen, Institute of Psychology, Goßlerstraße 14, 37073 Göttingen, Germany
[2]Department of Affective Neuroscience and Psychophysiology, University of Göttingen, Institute of Psychology, Goßlerstraße 14, 37073 Göttingen, Germany
[3]Leibniz ScienceCampus Primate Cognition, 37073 Göttingen, Germany
[4]Department of Developmental Psychology, University of Lüneburg, Germany

LK, 0000-0002-9696-8619

Recently, Theory of Mind (ToM) research has been revolutionized by new methods. Eye-tracking studies measuring subjects' looking times or anticipatory looking have suggested that implicit and automatic forms of ToM develop much earlier in ontogeny than traditionally assumed and continue to operate outside of subjects' awareness throughout the lifespan. However, the reliability of these implicit methods has recently been put into question by an increasing number of non-replications. What remains unclear from these accumulating non-replication findings, though, is whether they present true negatives (there is no robust phenomenon of automatic ToM) or false ones (automatic ToM is real but difficult to tap). In order to address these questions, the current study implemented conceptual replications of influential anticipatory looking ToM tasks with a new variation in the stimuli. In two separate preregistered studies, we used increasingly realistic stimuli and controlled for potential confounds. Even with these more realistic stimuli, previous results could not be replicated. Rather, the anticipatory looking pattern found here remained largely compatible with more parsimonious explanations. In conclusion, the reality and robustness of automatic ToM remains controversial.

## 1. Is implicit Theory of Mind real but hard to detect? Tests with different stimulus materials

Theory of Mind (ToM), the ability to attribute mental states to others, has traditionally been assumed to be a conscious form of

higher cognition that allows for the flexible ascription of subjective perspectives, depends on language and executive function and develops in protracted ways around the age of 4 years [1–3]. New research in the last 15 years, however, has challenged this traditional assumption. Various implicit measures suggest that even infants operate with basic ToM capacities and that automatic, spontaneous and non-conscious use of these capacities remains in operation throughout the lifespan [4–8]. These measures include *violation of expectation* looking time paradigms [4,5], *interactive behavioural tasks* [9–11], *priming* [12–14] and *anticipatory looking* measures (e.g. [6,8,15–18]). The latter have been used most widely across the lifespan [15], across species [19] and across typical and atypical populations [16,20].

These findings have drastically affected the field and have been taken as the evidential basis for far-reaching theoretical conclusions. *Nativist* accounts assume that these new implicit tasks tap the real, probably modular and innate core ToM capacities in the purest form. Traditional verbal studies, in contrast, have underestimated and masked early conceptual competence due to extraneous performance factors posed by the linguistic, working memory and other demands of the tasks (e.g. [21–23]). *Two-systems accounts of ToM* [24,25] assume that implicit tasks tap a more basic, evolutionarily and ontogenetically older *system I* of mindreading. This system differs in a number of crucial respects from the later developing *system II* of fully fledged, explicit and conscious mindreading. Both kinds of accounts thus agree that findings from the new implicit tasks document some early mindreading capacity (but differ with regard to the question whether this early capacity is already fully fledged ToM or some more basic capacity). Such far-reaching theoretical conclusions have been rejected by *skeptical accounts* which suggest that the findings from early implicit ToM tasks can be explained more parsimoniously as the product of 'sub-mentalizing', i.e. sensory and attentional processes [26–29].

For the most part, this debate between richer and more lenient accounts centers around the question how to interpret the findings from the implicit tasks on the premise that they constitute robust effects. This premise itself, however, has recently come under dispute. First of all, still relatively few published positive findings exist, several of which have small sample sizes (e.g. $n \leq 10$ per condition in Senju *et al.* [16,20] and Southgate *et al.* [17]). Furthermore, there is currently a growing body of non-replications of violation of expectation paradigms [30–32], interaction tasks [33,34] and anticipatory looking tasks (e.g. [35–39]). Furthermore, there are a considerable number of un-published non-replications in the file drawers around the world (see [40] for an overview). The rich conclusions thus seem to build on much less solid ground than previously assumed.

There are several possibilities why replication studies may have failed, including the following two: Firstly, non-replications might reflect true negatives, suggesting that there is no such thing as robust implicit ToM. Specifically, some studies show better replicability in older children and adults compared to infants, suggesting that explicit rather than implicit mechanisms may be in place [36,37,41]. Secondly, they may reflect false negatives in the sense that implicit ToM is real but difficult to tap; and the current paradigms may not have been sufficiently sensitive to measure implicit ToM. Recent research on non-human primates may be taken to support the second possibility: after decades of consistently negative findings from ToM tasks with non-human primates, a recent chimpanzee study documented some belief-tracking competence for the first time when the apes where confronted with more engaging and ecologically relevant stimuli [19]. Furthermore, high dropout rates in original (e.g. [16,17]) and replication studies [36] of up to 30% of the sample (because subjects did not engage in any anticipatory looking in familiarization trials) make the results of existing studies difficult to interpret.

The rationale of the present study was, therefore, to implement conceptual replications of existing anticipatory looking ToM tasks [17,18] with a focus on potential influences of the stimuli. Thus, tasks structurally analogous to those used by Southgate *et al.* [17] and Surian & Geraci [18] were used, but with more engaging stimuli. The crucial question was whether these modified stimuli can reduce dropouts and reveal, more sensitively, automatic belief tracking in adults. As a corollary aim, we attempted to increase the reliability of previous paradigms by removing potential confounds (for details, see below). Two studies implemented change-of-location false belief (FB) scenarios and combined, in a comprehensive design, conceptual replications of two types of tasks previously used: *Transfer* tasks are implicit adaptations of the explicit standard Wimmer & Perner [1] FB tasks [15,18,42]. An agent puts an object into box 1, which is then transferred to box 2, witnessed (true belief, TB condition) or unwitnessed (FB condition) by the agent, who then sets into motion to get the object. *Removal* tasks are simplified versions in which the target object is removed from the scene, and there is thus less inhibitory demand to suppress looking to the object's real location [16,17,20]: an agent puts an object into box 1. The object is then transferred to box 2, and finally removed from the scene. In the two conditions, the agent either witnesses steps 1 and 2, and thus believes the object to be in box 2 (FB1 condition), or

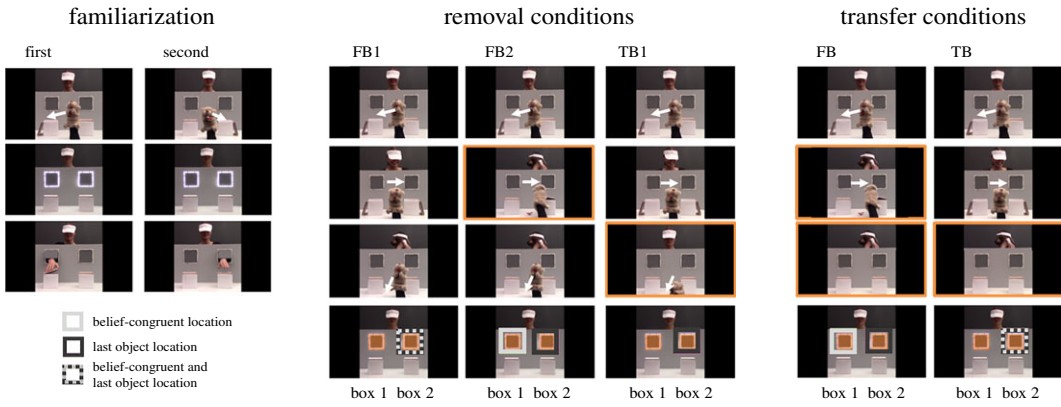

familiarization — removal conditions — transfer conditions

**Figure 1.** Screenshots from selected frames describing the different trials used in Study 1. Figures depict the two familiarization trials, the test trials of the removal conditions and the transfer conditions. Light grey boxes around the AOIs, depicted as orange rectangles, indicate the belief-congruent side and dark grey boxes the (last) object location. 'box 1' denotes the box into which the object was placed first, 'box 2' the one into which the object was then transferred. Note that the direction of the agent's turning was counterbalanced across trials.

witnesses only step 1, and thus believes it to be in box 1 (FB2 condition). In addition to the original removal trials (FB1 and FB2), a novel condition was introduced to control for the possibility that participants tracked the last object location rather than the agent's belief. In the novel true belief condition (TB1), the actress observes the full process (moving and removal of the object) and thus knows that the object has been removed from the scene (figure 1). If participants still were to look at the last object location in this condition in the same ways as in FB1, this would suggest that they possibly track the object's last location rather than anticipate where the actress will reach for the object in both conditions.

The present study tested with these modified procedures and stimuli whether previous findings could be conceptually replicated, supplying robust evidence for automatic ToM. The most clear-cut evidence for automatic ToM would be constituted by full conceptual replication of both Southgate *et al.* [17] and Surian & Geraci [18]: in the removal conditions, subjects look in anticipation as a function of the agent's belief both in FB1 and FB2, but do not show any anticipation in the new TB1 control condition (in which the agent knows that the object is no longer in the scene). In the transfer conditions, subjects look in anticipation as a function of the agent's belief both in FB (more anticipatory looking to the now empty box in which the agent falsely believes the target object to be) and in TB (more anticipatory looking to the object's actual location), or at the least they look differently to the two locations in FB and TB conditions. More fragile evidence could take the following form: subjects show belief-based anticipation in the removal conditions FB1 and FB2 (in which there is no conflict between the object's real location in the scene and the location where the agent believes it to be) but fail to do so in the more demanding transfer conditions FB/TB (in which there is such a conflict). Such patterns of results were recently found, for example, by Wang & Leslie [43].

Study 1 uses a general set-up and stimuli similar to the Southgate/Senju paradigm, with a controlled environment (figure 1) in which an agent reaches for objects through one of two windows. The main difference was that more engaging objects (chocolate) were used in order to increase the relevance of the object for the agent. Study 2 uses an even more vivid, engaging and emotionally laden scenario (figure 3). The removal conditions of both studies were preregistered with large predetermined sample sizes. In addition, the preregistration stated that effects might be followed up with additional transfer conditions based on the Surian & Geraci [18] paradigm, which was also included in this paper. Note that this was only preregistered as an exploratory part of the study.

# 2. Study 1

## 2.1. Methods

### 2.1.1. Participants

The current study was preregistered with the Open Science Framework (https://osf.io/hj9kr/). Based on Senju *et al.*'s study ([16]; see electronic supplementary material, A), the sample size was predetermined at

**Table 1.** Number of participants included and excluded in Study 1.

| participants | number |
|---|---|
| total | 217 |
| included | 125 |
| excluded | 92 |
| failed original Senju *et al.* [20] familiarization criterion | 58 |
| did not look at any AOI during the last familiarization | 14 |
| did not look at any AOI during the test trial | 17 |
| technical problems | 3 |

$N = 125$ ($n = 25$ in each of the three removal (FB1/FB2/TB1) and two transfer (FB/TB) conditions). To reach the required number of subjects, 217 adults were tested (age range 18–34 years, $M = 22.9$, s.d. = 2.9, 127 females), recruited on the campus of Göttingen University. From further analyses, 58 participants had to be excluded because they failed the inclusion criterion based on Senju *et al.* [20], meaning that they did not spend more time looking at the correct than at the incorrect any area of interest (AOI) during the last familiarization trial. Additional participants were excluded because they did not look at AOI during the crucial period in the last familiarization ($n = 14$; note that this case differs from the original Senju *et al.* [20] criterion, as no gaze was recorded at all and therefore no preference for looking to either AOI could be determined) and/or test trial ($n = 17$) or due to technical problems ($n = 3$) (table 1). All participants had normal or corrected to normal vision. They provided informed consent before taking part in the study. The study was approved by the University of Göttingen ethics committee and conducted in line with the Declaration of Helsinki.

### 2.1.2. Materials and stimuli

Stimulus presentation and recording of the participants' eye movements was controlled by a HP-computer (HP ZBook with a Windows 10 pro operating system) with a remote eye-tracker (RED 250 mobile) that sampled gaze positions at a rate of 60 Hz using the SMI iView X software (v. 3.6.53). Stimuli were generated on a 15.6″ display (39.58 cm, 1920 × 1080 pixel) using the SMI Experiment Center software (v. 3.6.53).

The stimuli of Study 1 were modelled as closely as possible on the original stimuli used in the Southgate/Senju paradigm. The original stimuli include familiarization trials followed by one of two test trials (FB1 or FB2). The Southgate/Senju paradigm uses the same, basic setting for all trials—a visored actress is standing behind a white panel. Two windows are set into the panel, each with an opaque box positioned in front of it. Participants are presented with a puppet bear moving a brightly coloured ball within the setting. The agent witnesses the puppet bear's actions until she gets distracted by a ringing telephone. After the phone stops ringing, the actress turns back. In order to elicit an anticipatory response to the agent's retrieval of the ball from one of the boxes, the two windows illuminate for one second accompanied by a chime after the agent has turned back. Participants were familiarized with these cues in preceding trials, with the visual and acoustic signal being followed by the agent reaching through one of the two windows to grab a toy after a 750 ms delay.

In the current study, to increase the relevance of the actions shown in the video, chocolate was used instead of the ball, and its importance was emphasized to the agent in a short, introductory video clip: a puppet bear appears while the agent enjoys a piece of chocolate. After discovering that no chocolate is left, the agent presents an empty bag to the puppet bear and expresses her unhappiness. The puppet bear seems to have an idea and leaves the scene. Subsequent to the introductory video clip, participants were presented with two familiarization trials (as in Southgate *et al.* [17]) and a test trial (see figure 1 for a schematic display of the different conditions and movie electronic supplementary material, B). In test trials, the outcome of the action is not revealed. Instead, following Senju *et al.* [16,20], participants were presented with a freeze frame for a 5-s period subsequent to the window illuminating and the chime sounding. The continuous ringing of the phone in the Southgate/Senju paradigm might distract participants and affect their implicit reasoning about the agent's beliefs [26]. Therefore, the agent answers the phone which consequently stops to ring in the current set of stimuli.

We furthermore attempted to increase the reliability of previous paradigms by removing potential confounds. The adapted stimuli controlled for the turning of the agent after the belief induction phase. In the original stimuli, the direction of the agent's turning is confounded with the belief-congruent location, meaning that the agent moves in the left direction in the FB1 condition (in which the object is located in the left box before being removed) and the other way around in the FB2 condition (with the object's last location being on the right). Previous studies suggest that the turn of the actress can significantly affect gaze patterns [39]. The current set of stimuli included each test trial with both types of turning movements counterbalanced across subjects. In summary, we controlled for [1] potential cueing by the actresses' turn and [2] ball tracking as an alternative explanation (through the TB1 condition).

### 2.1.3. Procedure

After signing a document of informed consent, participants were pseudo-randomly assigned to one of the conditions (FB1, FB2, TB1, FB or TB) based on a predetermined randomized list of the conditions. Allowing for the best possible eye-tracking signal, participants were seated at a distance and angle to the screen that was adjusted to their height. Before the stimulus presentation started, a standard five-point calibration was completed.

### 2.1.4. Statistical analyses

For the removal conditions (FB1, FB2, TB1), following the original study by Southgate *et al.* [17], anticipatory looking responses towards two AOIs were coded: the 'box 1' AOI included the windows corresponding to box 1 and the 'box 2' AOI included the window corresponding to box 2. In FB1, box 2 corresponds to both the belief-congruent location and the location the object was in last before removal. In FB2, box 1 corresponds to the belief-congruent location whereas box 2 corresponds to the last object location. There is no belief-congruent location in TB1, but like in FB1 and FB2, box 2 corresponds to the last object location. Looking times towards the 4.2 cm × 4.2 cm AOIs were analysed using BeGaze software (v. 3.6.53).

Following the original studies replicated here, two measures of anticipatory looking were used, the direction of first saccades after the illumination of the windows, and gaze duration over a 6-s period starting from the onset of window illumination. Binomial tests were used to investigate whether first saccades were directed systematically towards one of the AOIs and Fisher's exact tests were used to analyse whether participants directed their first saccades towards the two AOIs different between conditions.

In order to analyse whether looking times to the two AOIs differed from chance, and differed between conditions, differential looking scores (DLS) were computed: for FB1 and FB2, the DLS was calculated by dividing the looking time towards the belief-incongruent window subtracted from the looking time towards the belief-congruent window by the sum of looking times to both windows:

$$DLS_{belief} = \frac{LT_{bc} - LT_{bi}}{(LT_{bc} + LT_{bi})}.$$

This DLS can range from −1 to 1, with 1 reflecting increased looking durations towards the belief-congruent window, −1 reflecting increased looking durations towards the belief-incongruent window and 0 reflecting looking durations to the windows at chance level.

In order to control for low-level explanations in terms of merely looking to the last object location, FB1 and TB1 were compared. To this end, DLS for looking to the AOI corresponding to the last object location (box 2) in relation to overall looking at both locations were computed

$$DLS_{last\ object} = \frac{LT_{lol} - LT_{other}}{(LT_{lol} + LT_{other})}.$$

This DLS can range from −1 to 1, with 1 reflecting increased looking durations towards the AOI corresponding to the last location of the object (box 2), −1 reflecting increased looking durations towards the other AOI (box 1), and 0 reflecting looking durations to both AOIs at chance level. For FB1, since the 'box 2' AOI is both the belief-congruent AOI and the AOI corresponding to the last object location, this DLS is identical to the one above.

Whether DLS differed significantly from 0 was analysed using *t*-tests, and planned *t*-tests were conducted to compare DLS between conditions. The analyses were based on the preregistration of the

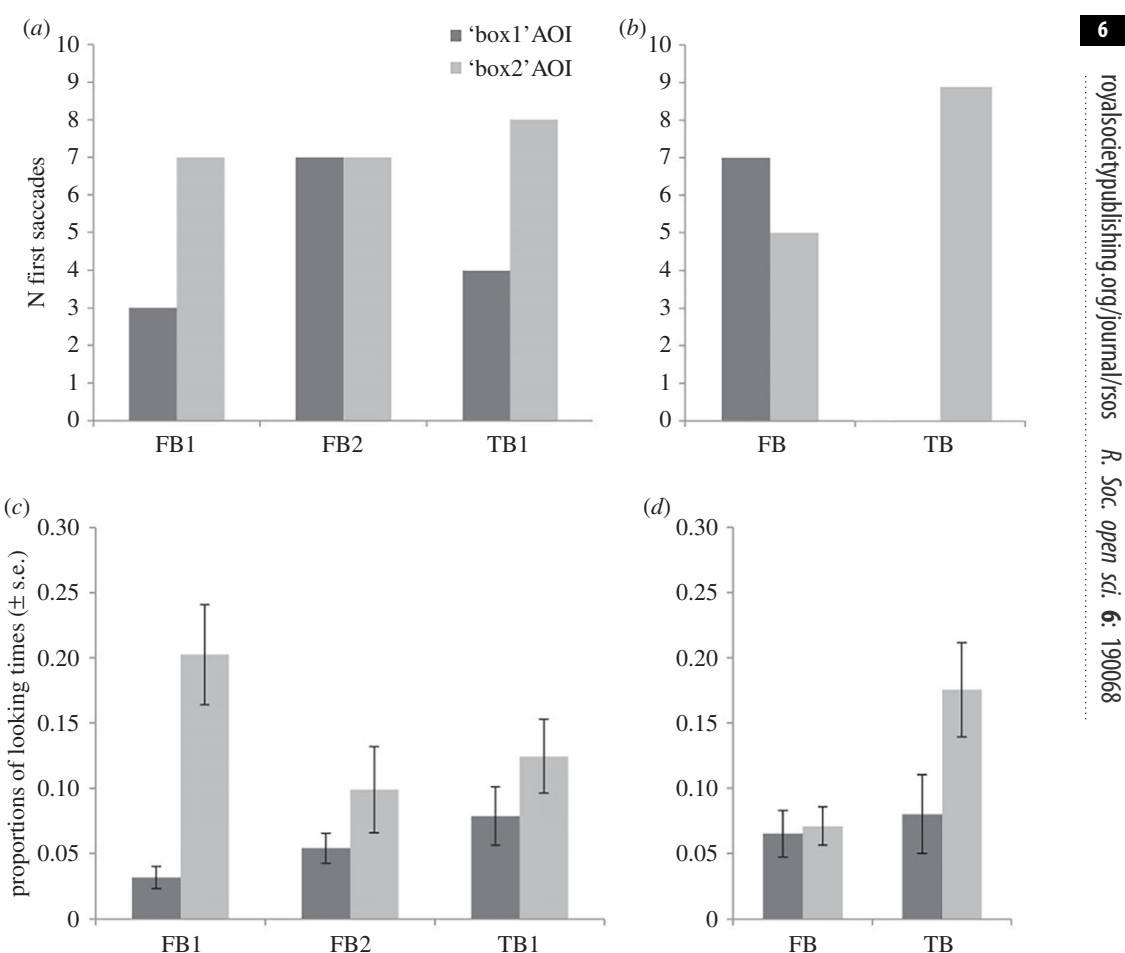

**Figure 2.** Number of first saccades to the correct and incorrect location (*a*,*b*), and proportional looking times for the removal (*c*) and transfer (*d*) conditions.

current study unless otherwise noted (see https://osf.io/hj9kr/ for details). Bayesian statistics evaluating the odds ratio of the null and alternative hypothesis were used to follow up potential null-effects in the preregistered tests. Bayes Factors were calculated using the proportionBF and the contingencyTableBF function, respectively, for first saccades and the ttestBF function regarding DLS.

Similarly, for the transfer conditions (FB and TB), two measures were employed for the analyses of anticipatory looking responses. First, the direction of first saccades was coded after the offset of illumination. Binomial tests were used to investigate whether first saccades were directed significantly more often towards the belief-congruent window (corresponding to box 2 in TB and to box 1 in FB) and Fisher's exact tests to test whether this effect differed between conditions (FB/TB). Second, proportional anticipatory looking times towards the two windows were analysed over a 6-s period starting from the onset of window illumination. The main analysis, following Surian & Geraci [18] and Schneider *et al.* [15], was a 2 (Condition: FB/TB) × 2 (AOI: box 1 versus box 2) ANOVA. DLS analyses of the transfer conditions are reported in the electronic supplementary material.

All statistical tests were carried out using RStudio v. 1.0.143 (R scripts for all analyses are provided in electronic supplementary material, C; the full dataset of the study is provided in electronic supplementary material, D). Bayesian analyses were conducted using the 'Bayes Factor' Package [44] in RStudio [45] using Cauchy priors based on Liang *et al.* [46].

## 2.2. Results

First saccades and proportional looking times in all conditions are displayed in figure 2. Note that descriptive statistics for all analyses, non-significant test results and Bayes factors are reported in the electronic supplementary material, E.

### 2.2.1. Removal conditions

*First saccades.* In all *removal* conditions (FB1, FB2 and TB1), participants directed their first saccades towards both windows at chance. Their contrasts did not reach significance (see electronic supplementary material, E, Table E.3).

*DLS: FB1 versus FB2.* The analysis revealed that, according to DLS in the *removal conditions*, participants looked significantly longer towards the belief-congruent window in FB1, $M_{DLS} = 0.673$, $t_{24} = 6.84$, $p < 0.001$, $d = 1.37$. In FB2, participants' looking duration towards the windows did not differ between the belief-congruent and incongruent window ($M_{DLS} = 0.103$). In comparison with FB2 participants spent a significantly higher proportion of time looking towards the belief-congruent window in the FB1 condition (FB1–FB2: $t_{43} = 4.50$, $p < 0.001$, $d = 1.27$).

*DLS: FB1 versus TB1.* In TB1, participants' looking duration did not differ between the two windows ($M_{DLS} = 0.214$). A comparison of FB1 and TB1 revealed that participants spent a significantly higher proportion of time looking towards the window corresponding to the last object location in the FB1 condition than in TB1, $t_{45} = 2.85$, $p = 0.007$, $d = 0.81$.

### 2.2.2. Transfer conditions

*Proportional looking time.* For an exploratory analysis of the *transfer conditions* TB and FB, a repeated measures general linear model was computed to investigate the effect of Condition (FB or TB), AOI (box 1 versus box 2) and the interaction term on proportions of looking time (non-preregistered). There was a significant effect of AOI, $F_{1,48} = 4.26$, $p = 0.045$, $\eta_p^2 = 0.081$ and Condition, $F_{1,48} = 4.64$, $p = 0.036$, $\eta_p^2 = 0.088$, but no significant interaction, $F_{1,48} = 3.33$, $p = 0.074$, $\eta_p^2 = 0.065$.

*First saccades.* Regarding the *transfer conditions*, all participants who did make any saccade ($n = 9$) directed their first saccades towards the belief-congruent window, $p = 0.004$, in the TB condition but they were looking at chance in the FB condition. Participants directed a significantly higher proportion of first saccades to the belief-congruent window in the TB condition than in the FB condition ($p = 0.007$).

## 2.3. Discussion

The main results of Study 1 were the following: in the removal conditions, findings were mixed and inconclusive. FB1 was partially replicated (in looking time, but not first fixation), but not FB2. However, the findings in FB1 seem not reducible to mere ball tracking since subjects did not equally look to the last ball location in TB1 as in FB1 (which are identical in terms of mere ball-tracking but differ with respect to the agent's belief). In the transfer conditions, participants showed more looking to the belief-congruent location only in TB, but not in FB. However, there was no significant interaction of looking to the object and empty location between the FB and TB condition, suggesting that there was no evidence that the pattern in TB differed from that observed in FB. Furthermore, the dropout rates due to participants not showing object tracking in the last familiarization trial were high. This suggests that the task may not be engaging participants sufficiently.

Study 1 used a fairly controlled environment similar to Southgate/Senju, while a recent implicit ToM study by Krupenye *et al.* [19] did find evidence of belief tracking using a more realistic chasing scenario. Therefore, the aim of Study 2 was to create even more realistic stimuli, with a realistic and emotional chase scenario similar to the one used by Krupenye *et al.* [19].

# 3. Study 2

## 3.1. Methods

### 3.1.1. Participants

The removal conditions of the current study were preregistered with the Open Science Framework (https://osf.io/65pv8/). All stimuli are shared at https://osf.io/2zp46/. The sample size was predetermined at 125 (25 per condition) based on Senju *et al.*s' study ([16]; see the electronic supplementary material, A). To reach the required number of subjects, 345 adults (age range 18–40 years, $M = 22.9$, s.d. $= 3.7$, 177 females) were recruited on the campus of Göttingen University. Forty-three participants had to be excluded from further analyses because they failed the inclusion

**Table 2.** Number of participants included and excluded in Study 2.

| participants | number |
| --- | --- |
| total | 345 |
| included | 125 |
| excluded | 220 |
| failed original Senju et al. [20] familiarization criterion | 43 |
| did not look at any AOI during the last familiarization | 118 |
| did not look at any AOI during the test trial | 50 |
| technical problems | 3 |
| experimenter error | 6 |

criterion based on Senju et al. [20]; additional participants were excluded because they did not look at the AOI during the crucial period in the last familiarization ($n = 118$) and/or test trial ($n = 50$), due to technical problems ($n = 3$) or experimenter errors ($n = 6$) (table 2). All participants had normal or corrected to normal vision. The study was approved by the University of Göttingen ethics committee.

### 3.1.2. Materials and stimuli

For stimulus presentation and recording of participants' eye movements, the same technical equipment as in Study 1 was used. In order to achieve ecologically more valid stimuli, the following methodological changes were applied to the original Southgate/Senju stimuli. (1) Similar to Study 1, a chocolate bar was used instead of a simple ball. (2) The agent did not wear a visor. Thus, her eyes were fully visible which, along with clear facial expressions, aimed to enhance her function as an agent and increase her salience. (3) A new set-up and the introduction of a second actress instead of a puppet bear made the stimuli more realistic. (4) In the original stimuli, the agent turns around attending to a ringing telephone during the belief induction phase. However, the head-turning and phone-ringing might distract participants and could depict an alternative explanation for alleged findings of automatic belief processing [26,28]. Accordingly, Senju et al. [16] found that participants with Asperger's syndrome tracked the agent's head movements less than neurotypical adults which might have resulted in the negative findings in the former sample population. Thus, adapted from Krupenye et al. [19], the agent's absence and presence was used as a belief induction variable in our new set of stimuli. (5) As in Study 1, three additional conditions, TB1, TB and FB were included.

The general set-up of the stimuli was the same for all videos: the agent (an actress) is standing in front of a grey curtain between two opaque boxes. At the beginning of each trial, a second actress appears from the foreground and places a chocolate bar in one of the boxes (see figure 3 and movie electronic supplementary material, F). In two familiarization trials (as in Southgate et al. [17]), participants watched the actress place the chocolate bar in the left and right of the two boxes, respectively, while the agent is watching. The agent later retrieves the chocolate bar from that box. In order to elicit the anticipatory response to the agent's retrieval of the chocolate, a doorbell sounds after the chocolate has been hidden and the agent leaves the scene. Preceding the agent's central reappearance and stop between the boxes, the picture blurs, momentarily. On her return she rubs her hands followed by a 1-s illumination of the boxes. After a delay of 1750 ms, she then reaches into one of the boxes. In test trials, the outcome of the action was not revealed. Instead, participants were presented with a freeze frame for a 5-s period following Senju et al. [16,20]. Whether the last location of the chocolate was the right or the left box during test trials was counterbalanced across subjects.

### 3.1.3. Procedure

After signing a document of informed consent, participants were pseudo-randomly assigned to one of the conditions (FB1, FB2, TB1, TB or FB) based on a predetermined randomised list of the conditions. Allowing for the best possible eye-tracking signal, participants were seated at a distance and angle to the screen that was adjusted to their height. Before the stimulus presentation started, a standard five-point calibration and validation routine was completed. Following the anticipatory looking task, participants completed a debriefing procedure (based on Schneider et al., [15] with minor alterations)

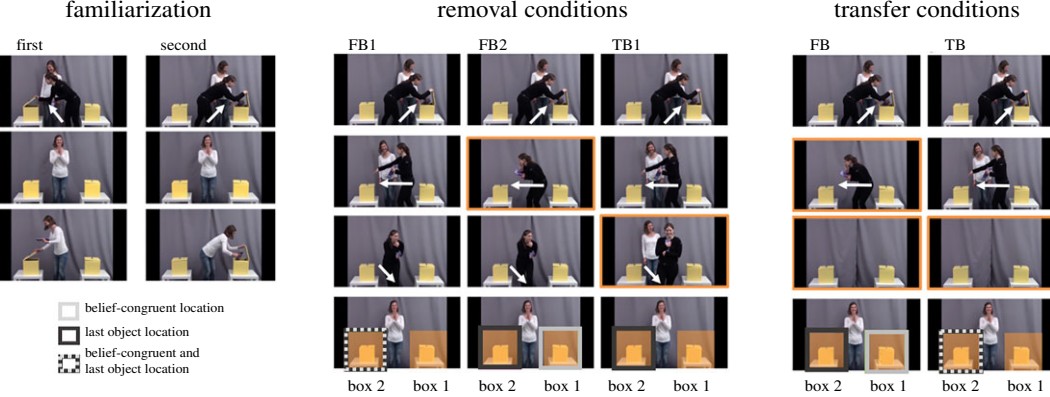

**Figure 3.** Screenshots from selected frames describing the different trials used in Study 1. Figures depict the two familiarization trials, the test trials of the removal conditions and the transfer conditions. Light grey boxes around the AOIs, depicted as orange rectangles, indicate the belief-congruent side and dark gray boxes the (last) object location. 'box 1' denotes the box into which the object was placed first, 'box 2' the one into which the object was then transferred. Note that the position of the boxes (whether box 1 was left or right) were counterbalanced across trials.

to investigate whether they guessed the aim of this more realistic setting, leading to a less distinct measure of implicit ToM. Furthermore, after the debriefing, participants watched the last sequence of the video again and were asked 'Where will the actress look for the chocolate?' as a verbal measurement of the participants' explicit FB understanding (see electronic supplementary material, G).

### 3.1.4. Statistical analyses

Like in Study 1, first saccades and DLS were used as measures of participants' anticipatory looking, with DLS being calculated from looking times towards the 6.2 cm × 5.5 cm AOIs that included the boxes (figure 3). Transfer conditions (TB and FB) were further analysed in regards to proportions of looking time (exploratory analysis) and DLS analyses are reported in the electronic supplementary material. Participants' responses to the explicit control question were coded as incongruent or congruent with the agent's belief. Whether participants passed the explicit control question above chance was tested using binomial tests (electronic supplementary material, G).

For additional analyses of dropout rates, those participants were considered who displayed longer looks towards the incorrect box on the last familiarization trial (adopted from Senju *et al*. [20] and Southgate *et al*. [17]) in proportion to the number of participants who passed the original inclusion criterion. The current dropout rate was compared with the dropout rates reported by Southgate *et al*. [17] as well as Senju *et al*. [20] and in Study 1 of the current paper, using binomial tests. The analyses were based on the preregistration of the current study (see https://osf.io/65pv8/ for details) unless otherwise noted.

## 3.2. Results

First saccades and proportional looking times in all conditions are displayed in figure 4. Note that descriptive statistics for all analyses, non-significant test results and Bayes factors are reported in electronic supplementary material, E.

### 3.2.1. Removal conditions

*First saccades.* In all *removal conditions* (FB1, FB2, TB1), participants directed their first saccades towards the boxes at chance. The contrasts between conditions did not reach significance.

*DLS.* In all *removal conditions*, participants' looking duration towards the boxes measured through DLS did not differ between the boxes. Participants' performance did not differ significantly between the removal conditions.

*Dropout rates.* In order to analyse dropout rates, only the number of participants who were excluded for not passing the original Southgate/Senju criterion of correct anticipation in the last familiarization trial and the number of participants who were included was considered. The current dropout rate

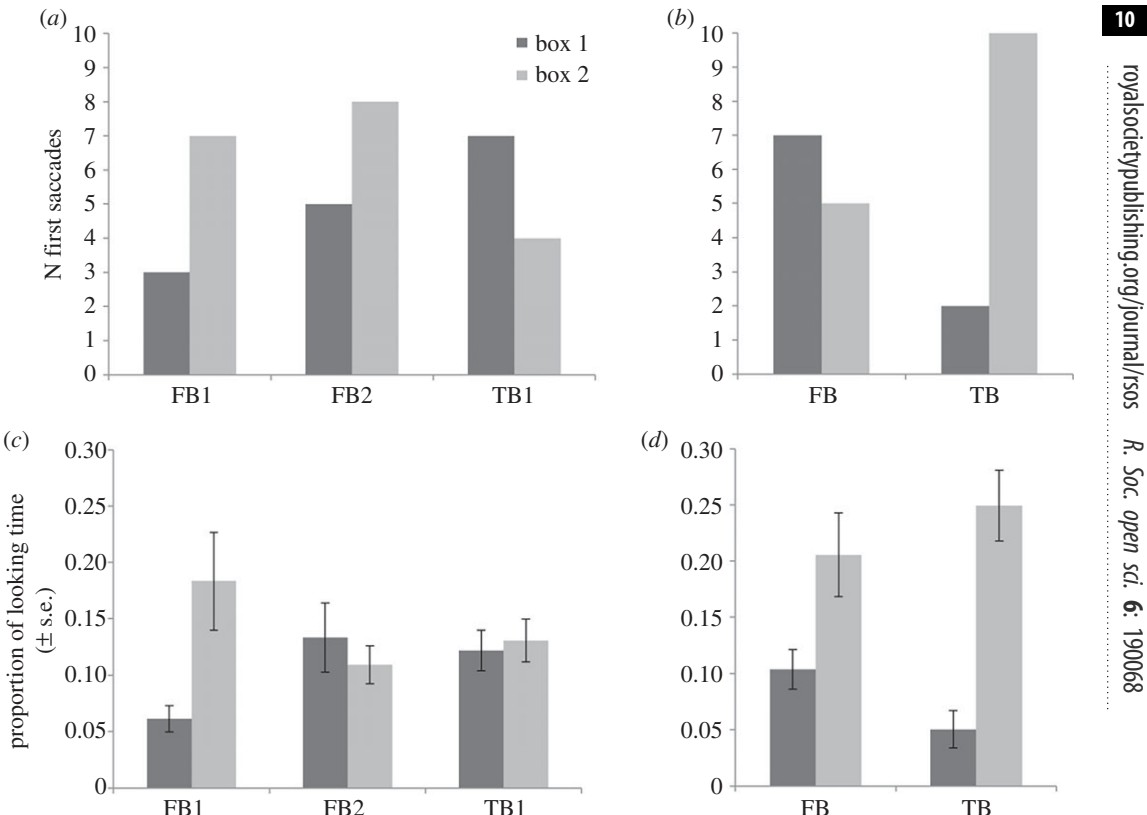

**Figure 4.** Number of first saccades to the correct and incorrect location (*a,b*) and proportional looking times for the removal (*c*) and transfer (*d*) conditions.

differed significantly from those reported in previous studies (current study: 26% (125 participants were included, and 43 failed the familiarisation criterion), Southgate *et al.* [17]: 35%, difference: $p = 0.008$; Senju *et al.* [20]: 11%, difference: $p < 0.001$). While the percentage of participants that failed the inclusion criterion was smaller than in Southgate *et al.*'s paradigm [17], the percentage increased in comparison with the dropout rate reported by Senju *et al.* [20]. There was no difference between the current study and Study 1 (Study 1: 32% (125 participants were included, and 58 failed the familiarisation criterion), $p = 0.09$).

### 3.2.2. Transfer conditions (exploratory)

*Proportions of looking time.* A mixed ANOVA (repeated measures general linear model; non-preregistered) computing the effects of AOI (box 1 or box 2), Condition (TB or FB) and their interaction on proportions of looking time showed a significant effect of AOI, $F_{1,48} = 27.07$, $p < 0.001$, $\eta_p^2 = 0.361$ but no significant interaction, $F_{1,48} = 2.84$, $p = 0.099$, $\eta_p^2 = 0.056$, and no main effect of Condition, $F_{1,48} = 0.04$, $p = 0.850$, $\eta_p^2 = 0.001$.

*First saccade.* In the *transfer conditions*, participants showed significantly more first saccades towards the belief-congruent box, ($n = 10$) than towards the other box ($n = 2$) in the TB condition, $p = 0.039$, but were looking at chance in the FB condition. The contrasts between the conditions did not reach significance.

## 3.3. Analysis of DLS across both studies

An exploratory analysis was performed to analyse DLS across the combined dataset of Study 1 and Study 2. As in both Studies 1 and 2, DLS was significantly positive in the TB condition ($M = 0.638$, $95\%CI = [0.491; 0.785]$), $t_{49} = 8.745$, $p < 0.001$, and did not differ from zero in the FB2 condition ($M = -0.041$, $95\%CI = [-0.229; 0.146]$), $t_{49} = -0.444$, $p = 0.659$, or the FB condition ($M = -0.037$, $95\%CI = [-0.253; 0.180]$), $t_{49} = -0.341$, $p = 0.735$. The combined DLS in the FB1 condition was now significantly

positive ($M = 0.477$, $95\%CI = [0.300; 0.654]$), $t_{49} = 5.429$, $p < 0.001$, as previously only in Study 1 but not Study 2. Furthermore, there was now a marginally significant effect in the TB1 condition that was previously not significant in either study ($M = 0.173$, $95\%CI = [0.012; 0.333]$), $t_{49} = 2.165$, $p = 0.035$. For detailed analyses, see electronic supplementary material, H.

## 4. General discussion

The main aim of the current study was to conceptually replicate anticipatory looking ToM measures with more engaging stimuli that have the potential to tap at implicit ToM more sensitively, and with better controlled procedures that exclude previous confounds and rule out alternative explanations. The main results were the following: In the *removal conditions* based on Southgate *et al.* [17] (FB1, FB2, TB1), first saccades were directed towards both locations at chance and did not differ between conditions. In terms of looking times (indicated in DLS), anticipatory looking was directed to the belief-congruent location at above-chance levels in the FB1 condition, but only in Study 1 and the combined data of both studies, but not in Study 2. For all other conditions, DLS was at chance. Regarding the new control condition TB1, anticipatory looking (indicated in DLS) differed between FB1 and the TB1 condition, but again only in Study 1 and not in Study 2. Note however, that when combining both datasets, DLS was significantly positive in this condition, indicating object tracking.

In the *transfer conditions*, first saccades were directed at chance in the FB condition and above chance to the correct location in the TB condition in both studies, with the difference between conditions being significant in Study 1 only. Proportions of looking time merely showed a significant effect of location such that subjects looked longer to the object than the empty location, but there was no effect of belief condition (FB/TB) and particularly no interaction effect between belief condition and location. DLS consistently differed from zero in the TB condition in both studies as well as the combined dataset, but did not differ from zero in the FB condition.

In terms of replication, the present findings thus only partially replicate the results of previous studies. In the removal conditions, only the FB1 condition was significantly replicated in Study 1, while the more controlled FB2 condition was at chance. However, effects in FB1 seem not reducible to mere belief tracking in Study 1 which revealed a difference in anticipatory looking between the FB1 and the new TB1 condition that was explicitly designed to control for object tracking. This difference, though, was not found in Study 2, and so overall it remains still unclear whether object tracking is an alternative explanation. In the transfer conditions, there was no replication of the main findings (i.e. the interaction effect due to increased looking to the empty location in the FB condition). Only the TB but not the FB condition was replicated, which was originally designed as a baseline condition and is therefore not informative by itself.

In summary, only those conditions could be replicated where the belief-congruent location was identical to the (last) object location. And these are the conditions most obviously subject to alternative explanations. The current partial/non-replication converges with an increasing number of non-replications of implicit ToM tasks (for an overview, see [40]). In particular, it is in line with other recent results that only found looking behaviour in line with belief processing in those conditions that are prone to alternative explanations [33,34,36,37,39,40,47]. Note, however, that there was a slight improvement compared to previous failed replications such that anticipatory looking was at chance in the FB2 condition rather than being significantly more often directed to the belief-incongruent (i.e. object) location [36]. This, together with the results from the new control condition TB1 in Study 1, indicates that participants did not simply and indiscriminately look at the last object location. The current results may thus be the first to suggest that perhaps also for humans more engaging stimuli are more sensitive to uncover automatic belief tracking. Nevertheless, potential effects are clearly very small and more systematic future research is needed. In Study 2, the agent's absence from a scene during a belief induction may actually make it more difficult to track her perspective than simply turning around (e.g. [48]). However, the disappearing of the actress was in line with the study by Krupenye *et al.* [19], which the stimuli were based on and which found significant belief-tracking effects despite the disappearance of the actor. It should further be noted that recent research confirms that adults are able to correctly interpret FB scenarios, for example, in violation of expectation tasks [49], suggesting that the lack of belief-tracking behaviour is not related to a lack of skills but rather to the unsuitability of the task. In specific, the first anticipatory looking FB task by Clements & Perner [6] was found to be reliably replicable [40], possibly because it is easier for participants to follow [50]. The design of the task may therefore play a crucial role for anticipatory looking measures.

Overall, the present studies thus remain inconclusive—despite the large sample sizes. On the one hand, the most unambiguous and stringent conditions (FB2 and FB) did not yield any convincing evidence for belief-ascription. On the other hand, the less stringent conditions (FB1 and TB) did produce some positive evidence for anticipatory looking consistent with belief tracking. These patterns of anticipatory looking taken by themselves remain difficult to interpret given the ambiguous nature of the conditions, but they do not seem to merely reflect tracking of last object location, as indicated in the new TB1 control condition.

In addition, it needs to be noted that dropout rates remained high despite the more engaging videos. This is in contrast to the original aim of the study to reduce dropout rates. As dropout rates are in line with other recent implicit ToM studies, some of which were using the original stimuli by Southgate *et al.* [39,41,51], the high dropout rates do not seem related to the stimulus material but rather to the task. Note that the dropout rate calculations were based on those participants who did not display any object tracking and whose gaze therefore did not indicate that they were following the plot. They are therefore not related to procedural errors but to the measure of the task (anticipatory looking) not indicating any plot following. There are two possible explanations for this finding. Firstly, participants may be unable to follow the plot. As adults were tested who are perfectly able to solve explicit FB tasks and who are able to follow FB scenarios in violation of expectation tasks [49], this would not be due to a lack of skills but rather due to the task being unsuitable. Secondly, the outcome measure (anticipatory looking) may not be reliable for measuring ToM. If correct anticipatory looking does not even occur in the simple familiarization condition, this questions the overall suitability of anticipatory looking measures for measuring ToM, both in the original stimulus materials and in the ones developed for the current study.

One aim of the current set of studies was to develop more engaging stimulus materials. For this purpose, a step-wise approach was chosen, increasing relevance to the participants in Study 1 by using more interesting objects (chocolate, which participants enjoyed eating) and by using a more realistic chasing scenario based on Krupenye *et al.* [19] in Study 2. The changes did not lead to a reliable improvement in belief tracking. By contrast, the findings were in line with other non-replications using the original stimuli [36,39]. The anticipation performance during familiarization trials was also not improved in the current study compared to previous replications, but was on a similar level as in the paper by Krupenye *et al.* [19]; therefore, this performance may be unrelated to the engagingness of the stimuli. In summary, the kind of task does not seem to reliably work, irrespective of the specific stimulus materials that were used. This seems to point towards fundamental problems with the paradigm as such, rather than raise challenges related to the present stimulus materials.

This inconclusive situation calls for more systematic, future research. In particular, future studies need to tap different potential factors underlying anticipatory looking behaviour (belief tracking versus object tracking) in more nuanced and fine-grained quantitative ways in systematic factorial designs. More generally, the partial pattern of replications using more realistic stimuli leave open two broad possibilities. They might suggest that automatic ToM is real but fragile and difficult to tap. True mindreading processes might either be masked by other factors (e.g. salience of objects in the scene) or fine-grained technicalities in anticipatory looking tasks (e.g. speed of the video, see [19]), or anticipatory looking might simply not be the right kind of measure to tap automatic ToM processes. Alternatively, these tasks could be sensitive measures of some form of implicit mindreading—but one that is qualitatively different from and significantly less complex than proper belief-ascription [24,52,53]. In the current and previous studies, the difference between FB and TB has been confounded with a difference in knowledge/information access versus ignorance/lack of information. In the TB condition, the agent watches the scene and participants might thus attribute knowledge to her, while in the more complex FB condition she does not observe all the crucial events and participants might thus simply attribute ignorance. Participants may expect the knowing agent to act successfully and go to the real location of the object in the TB condition. In the FB condition, they simply hold the agent ignorant (or even fail to form any representation concerning her epistemic status; [54]) and thus have no firm expectation where she would go. This way of tracking the agent's knowledge versus ignorance, though conceptually much simpler than proper belief tracking, would be sufficient to yield the anticipatory looking patterns found here. And such an explanation would fit with other recent patterns of replications and non-replication results across implicit tasks with varying demands and complexity. In a study by Powell *et al.* [32], for example, those tasks that strictly require FB understanding proved not to be robust, whereas those that can be solved by distinguishing knowledge and ignorance could be reproduced. Again, more systematic investigations in the future are required to disentangle proper belief-tracking from simpler social-cognitive processes.

# 5. Conclusion

In summary, the use of more engaging stimuli in anticipatory looking tasks in the current study led to slightly more nuanced anticipatory looking patterns compared to previous non-replication studies. However, the patterns of anticipatory looking found here still do not constitute stringent evidence for automatic belief tracking and remain ambiguous with regard to the underlying cognitive processes. In order to find out whether implicit ToM is real but difficult to detect, or whether its existence has been assumed on the basis of false positive results, future research needs systematic large-scale and multi-laboratory replication studies.

Ethics. The studies were approved by the University of Göttingen ethics committee (Ref. number: 143b) and conducted in line with the Declaration of Helsinki. Informed consent was obtained from all participants.

Data accessibility. Full datasets and scripts are provided in the supplements. The video files used as stimulus material can be downloaded from the OSF (Study 1: https://osf.io/rdb5k/, Study 2: https://osf.io/2zp46/).

Authors' contributions. L.K. was involved in the conception and design of the work, supervising the acquisition and data analysis, interpreting the data, drafting the manuscript and revising it critically for important intellectual content. M.W. was involved in the design of the work, the acquisition, analysis and interpretation of data and revising the manuscript. H.R. was involved in the conception and design of the work, supervising the data analysis, interpreting the data and revising the manuscript critically for important intellectual content.

Competing interests. The authors declare no conflict of interest.

Funding. We received no funding for this study.

Acknowledgements. We thank Isabel Ganter, Max Hinrichs, Beatrice Holzapfel, Clara Hums and Melanie Zylenas for their help with recruitment and testing of participants and Teresa Illner and Marika Reimer for their help with filming the stimulus materials.

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
