## [Reviewer comments · Royal Society Open Science]

Review History

RSOS-190068.R0 (Original submission)

Review form: Reviewer 1

Is the manuscript scientifically sound in its present form?

Yes

Are the interpretations and conclusions justified by the results?

Yes

Is the language acceptable?

Yes

Is it clear how to access all supporting data?

No

Do you have any ethical concerns with this paper?

No

Have you any concerns about statistical analyses in this paper?

Yes

Recommendation?

Accept with minor revision (please list in comments)

Comments to the Author(s)

This paper makes an important contribution to the heated discussion about the validity of the so-called implicit tests of false belief understanding at very young ages. The contribution is indirect by testing whether adults can make sense of stimuli to which infants are subjected. The results make clear that even adults often miss the intended interpretations of the presented stimuli.

The paper neglects two recent studies using very similar stimuli for adults. Both of them provide some positive evidence for translocation studies: Low & Edwards 2018 for violation of expectation and, directly relevant, Schuwerk et al 2018 for anticipatory looking (AL). The authors need to discuss how their mostly negative results relate to these weak but positive results and what we can conclude about the reliability of these effects.

Low, J., & Edwards, K. (2018). The curious case of adults' interpretations of violation-of-expectation false belief scenarios. *Cognitive Development*, 46, 86-96.

Schuwerk, T., Priewasser, B., Sodian, B., & Perner, J. (2018). The robustness and generalizability of findings on spontaneous false belief sensitivity: a replication attempt. *Royal Society open science*, 5(5), 172273.

In the presentation of results a clear distinction should be made between predicted effects by different hypotheses (looking at believed location, looking at last object location, looking at head-turn location, ...) permitting use of t-test. Fortuitous findings, e.g., looking time to "box 2" for FB1 > TB1, without including FB2 in the test, need to be a-posteriori corrected. Although the overall number of participants is quite large, the numbers per condition seem insufficient to get even massive looking effects significant (e.g., the difference in first looks between FB and TB in Study 2). Perhaps one could add an additional analysis for the data of both experiments to reduce the number of relevant looking but insignificant effects.

The confounding between head turn and belief-congruent looking was emphasized as an important improvement in the Introduction but the relevant data are not presented. It would be interesting to see whether this confounding does have an effect.

Minor points:

On page 3 we read "These findings have revolutionized the field," which sounds odd in the context of a paper that claims that the reported findings are still open to doubt. How could unreliable data have revolutionized a field?

The use of "clockwise" on page 9 is ambiguous. Presumably it refers to a bird's eye view of the agent's head turn.

The text, "all participants who did make any saccade (n = 9) directed their first saccades towards the belief-congruent window, p = .004, in the TB condition," on page 14 does not match Figure 2, where all 9 looked at "box 1," which according to Figure 1 is not the belief congruent window. Page 19: "pseudo-randomly assigned to one of the conditions (FB1, FB2, TB1, TB or FB) based on a predetermined randomised list of the conditions." Was this done once for all children or for each child individually?

Page 23: "In Study 2, the actor's absence from a scene during a belief induction may actually make it more difficult to track her perspective than simply turning around (see e.g., Rubio-Fernández & Geurts, 2013) However, the disappearing of the actress was in line with the study by Krupenye et al. (2016), which the stimuli were based on." Presumably the finding by Rubio-Fernandez & Geurts speaks against the use of this method in this study designed to make the tasks as easy as possible. But how does the fact that the study followed Krupenye et al alleviate this problem?

The verdict, "this questions the overall suitability of anticipatory looking measures for measuring Theory of Mind," on page 24 ignores the fact that the original method (Clements & Perner 1994)

of measuring AL in the traditional FB task, with text and not just based on inference from visual observations, seems to have produced highly reliable evidence of earlier understanding than answers to questions (Kulke & Rakoczy 2018). From this one might want to draw the alternative conclusion that in the non-verbal paradigms of Southgate et al children and even adults find it difficult to understand what the purpose of the agent's actions are. But once they understand this – because the story makes it clear – AL may well be a useful method.

Review form: Reviewer 2

Is the manuscript scientifically sound in its present form?

No

Are the interpretations and conclusions justified by the results?

No

Is the language acceptable?

Yes

Is it clear how to access all supporting data?

Yes

Do you have any ethical concerns with this paper?

No

Have you any concerns about statistical analyses in this paper?

No

Recommendation?

Reject

Comments to the Author(s)

Review: "Is implicit Theory of Mind real but hard to detect? Tests with different stimulus materials"

The authors report a failed replication of anticipatory-looking false-belief studies by Southgate et al. (2007) and Surian and Geraci (2012) with adult participants. The aim of this replication study was to investigate automatic Theory of Mind using improved, more engaging materials with adults. As in other recent replication attempts of Theory of Mind studies, the original results were not replicated. Study 1 reports increased looks to a belief-congruent rather than a belief-incongruent location in a simple FB condition and random looking in a TB condition controlling for a last-location bias. However, Study 2 (the allegedly more engaging and realistic study) fails to replicate either of these findings. Overall, both studies find more first saccades to the belief-congruent location in a TB but not in an FB condition.

Learning about failed replications can be valuable and important for a field, but in this particular case, it is not clear what we can learn from these results. There is not a single conclusion that could be unequivocally drawn from this study. The original findings were partially replicated in Study 1, but this partial replication was not observed in Study 2, making both sets of results uninterpretable. More generally, as the authors concede in the General Discussion, their failure to replicate the original results with allegedly improved methods cannot be taken as evidence that

there is not such a thing as automatic Theory of Mind. If that is the case, what can we learn from this study?

More specifically, I have three main issues with this manuscript: (1) the dropout rates are extremely high for a study with adults, (2) there is no evidence that the stimuli are better or any more 'engaging' or 'realistic' than those used in previous studies, (3) trying to publishing null results when one's experimental manipulations have clearly failed is generally questionable.

1) The dropout rates in this study are abnormally high for a study with adults. Almost half the sample recruited in Study 1 and almost 2/3 in Study 2 were eliminated from analyses. The authors fail to discuss dropout rates in Study 1, and seem to make a mistake when reporting dropout rates in Study 2: on p.20 (line 43) the dropout rate is reported at 26%, but in the Participants section (p.16) it is reported at 220 out of 345 adults, which is 64% of the total sample.

In Study 2, the authors compare their dropout rates to the original studies with infants (Southgate et al. 2007) and children (Senju et al. 2010) rather than the study that was conducted with adults (Senju et al. 2009). This is very puzzling since it was this latter study on which the sample size for the study was based (reasonably, since both studies employed an adult population). More importantly, dropout rates normally differ between studies with infants and children, on the one hand, and studies with adults, on the other, so running a study with adults and having similar dropout rates to studies with infants and children strongly suggests that there is something wrong with the paradigm. Senju et al. (2009) report no neurotypical adults being excluded for failing to anticipate in their familiarization trials, and they had designed their trials to avoid this.

If the majority of adults didn't anticipate the agent's course of action in a straightforward familiarization trial, this basic result calls into question the validity of the experimental procedure. The authors themselves seem to agree with this view, saying early in the paper that "high dropout rates... make the results of existing studies difficult to interpret" (p.5). However, while the authors are ready to criticize the original studies for these methodological issues, when interpreting their own findings in the General Discussion, they do not give their high dropout rates any consideration.

2) The pitch of this replication study is that the stimuli were more 'engaging', 'realistic' and 'ecologically valid' than those used in the original studies. However, the only change in Study 1 was to use a chocolate bar instead of a ball (not entirely clear why that should be an improvement, especially with adults). The changes in Study 2 were also pretty minimal: the actor didn't wear a visor, she stood up and moved around, and left the scene than turning her head. The Krupenye paper that the authors refer to was much more convincingly 'engaging' since it had a natural looking environment where an actor was trying to hit another actor with a big stick dressed up as a monkey.

If anything, it seems like the scenes in this study were not engaging enough. The fact that most adults in both studies were unable to anticipate the agent's goal in the familiarization trials suggests this very strongly. Given the importance of increasing participants' engagement in the task for the aim of the study, it seems clear that the authors should have piloted their stimuli and compared them with those used in the original studies. Without such a comparison (e.g., adults explicitly found these stimuli more engaging, watched them more attentively, or found them easier to follow), the conclusion to be derived in view of the present results is that the authors failed in their attempt to create better materials. It must be noted, however, that this is a failure of the present study, and not a failure to replicate previous work.

3. Related to the last point, we should consider what would have happened if this study had been an original study, and not a conceptual replication of previous work. In view of their null results,

the authors would have had to admit – as most of us have had to admit often enough – that their study had failed: the manipulations they had carefully introduced to tap a certain effect simply didn't work. However, while researchers addressing new questions with new paradigms face the risk of failing, researchers only aiming to replicate previous studies seem to have found a new business model in Academia: if after a few failed replications, researchers finally manage to replicate the original results, the replicated results would be news and therefore publishable material. However, if their manipulations failed (as they clearly did here), they can always write up a new failed replication and continue to question previous findings, while not adding anything new to the literature.

I think this practice is highly questionable as it makes failed replications immune to failure (ironically) and always publishable, regardless of the possible shortcomings that would prevent publication of original studies. Since publication standards should be just as high for original work and replication studies, I cannot recommend this manuscript for publication.

Decision letter (RSOS-190068.R0)

15-Mar-2019

Dear Dr Kulke,

The editors assigned to your paper ("Is implicit Theory of Mind real but hard to detect? Testing adults with different stimulus materials") have now received comments from reviewers. We would like you to revise your paper in accordance with the referee and Associate Editor suggestions which can be found below (not including confidential reports to the Editor). Please note this decision does not guarantee eventual acceptance.

Please submit a copy of your revised paper before 07-Apr-2019. Please note that the revision deadline will expire at 00.00am on this date. If we do not hear from you within this time then it will be assumed that the paper has been withdrawn. In exceptional circumstances, extensions may be possible if agreed with the Editorial Office in advance. We do not allow multiple rounds of revision so we urge you to make every effort to fully address all of the comments at this stage. If deemed necessary by the Editors, your manuscript will be sent back to one or more of the original reviewers for assessment. If the original reviewers are not available, we may invite new reviewers.

- Data accessibility

<http://datadryad.org/submit?journalID=RSOS&manu=RSOS-190068>

- Competing interests

- Authors' contributions

- Acknowledgements

- Funding statement

on behalf of Dr Antonia Hamilton (Associate Editor) and Professor Essi Viding (Subject Editor)
 openscience@royalsociety.org

Associate Editor's comments (Dr Antonia Hamilton):

The reviewers were very divided on this paper, with one recommending accept and the other recommending reject. In this case, a critical factor is whether the current study closely matches the pre-registration or not. Unfortunately, when I looked on OSF I found only the stimuli. I did not find a protocol for the study, a list of the criteria for including / excluding participants or trials, a plan of the analysis methods or any of the other things that I would expect to see in a full pre-registration.

I know the OSF site can be tricky, but it is important that these pre-reg files are publicly available and can be checked.

If you would like to revise this paper, please include links to the full pre-reg information and include a paragraph or two in the discussion to deal with the issues raised by reviewer 2. In particular, the large number of exclusions is potentially an issue in arguing that the stimuli are more engaging.

If a more detailed pre-reg document is not available, the paper will have to be rejected.

Comments to Author:

Reviewers' Comments to Author:

Reviewer: 1

Comments to the Author(s)

This paper makes an important contribution to the heated discussion about the validity of the so-called implicit tests of false belief understanding at very young ages. The contribution is indirect by testing whether adults can make sense of stimuli to which infants are subjected. The results make clear that even adults often miss the intended interpretations of the presented stimuli. The paper neglects two recent studies using very similar stimuli for adults. Both of them provide some positive evidence for translocation studies: Low & Edwards 2018 for violation of expectation and, directly relevant, Schuwerk et al 2018 for anticipatory looking (AL). The authors need to discuss how their mostly negative results relate to these weak but positive results and what we can conclude about the reliability of these effects.

Low, J., & Edwards, K. (2018). The curious case of adults' interpretations of violation-of-expectation false belief scenarios. *Cognitive Development*, 46, 86-96.

Schuwerk, T., Priewasser, B., Sodian, B., & Perner, J. (2018). The robustness and generalizability of findings on spontaneous false belief sensitivity: a replication attempt. *Royal Society open science*, 5(5), 172273.

In the presentation of results a clear distinction should be made between predicted effects by different hypotheses (looking at believed location, looking at last object location, looking at head-turn location, ...) permitting use of t-test. Fortuitous findings, e.g., looking time to "box 2" for FB1 > TB1, without including FB2 in the test, need to be a-posteriori corrected. Although the overall number of participants is quite large, the numbers per condition seem insufficient to get even massive looking effects significant (e.g., the difference in first looks between FB and TB in Study 2). Perhaps one could add an additional analysis for the data of both experiments to reduce the number of relevant looking but insignificant effects.

The confounding between head turn and belief-congruent looking was emphasized as an important improvement in the Introduction but the relevant data are not presented. It would be interesting to see whether this confounding does have an effect.

Minor points:

On page 3 we read "These findings have revolutionized the field," which sounds odd in the context of a paper that claims that the reported findings are still open to doubt. How could unreliable data have revolutionized a field?

The use of "clockwise" on page 9 is ambiguous. Presumably it refers to a bird's eye view of the agent's head turn.

The text, "all participants who did make any saccade ($n = 9$) directed their first saccades towards the belief-congruent window, $p = .004$, in the TB condition," on page 14 does not match Figure 2, where all 9 looked at "box 1," which according to Figure 1 is not the belief congruent window.

Page 19: "pseudo-randomly assigned to one of the conditions (FB1, FB2, TB1, TB or FB) based on a predetermined randomised list of the conditions." Was this done once for all children or for each child individually?

Page 23: "In Study 2, the actor's absence from a scene during a belief induction may actually make it more difficult to track her perspective than simply turning around (see e.g., Rubio-Fernández & Geurts, 2013) However, the disappearing of the actress was in line with the study by Krupenye et al. (2016), which the stimuli were based on." Presumably the finding by Rubio-Fernandez & Geurts speaks against the use of this method in this study designed to make the tasks as easy as possible. But how does the fact that the study followed Krupenye et al alleviate this problem?

The verdict, "this questions the overall suitability of anticipatory looking measures for measuring Theory of Mind," on page 24 ignores the fact that the original method (Clements & Perner 1994) of measuring AL in the traditional FB task, with text and not just based on inference from visual observations, seems to have produced highly reliable evidence of earlier understanding than answers to questions (Kulke & Rakoczy 2018). From this one might want to draw the alternative conclusion that in the non-verbal paradigms of Southgate et al children and even adults find it difficult to understand what the purpose of the agent's actions are. But once they understand this – because the story makes it clear – AL may well be a useful method.

Reviewer: 2

Comments to the Author(s)

Review: "Is implicit Theory of Mind real but hard to detect? Tests with different stimulus materials"

The authors report a failed replication of anticipatory-looking false-belief studies by Southgate et al. (2007) and Surian and Geraci (2012) with adult participants. The aim of this replication study was to investigate automatic Theory of Mind using improved, more engaging materials with adults. As in other recent replication attempts of Theory of Mind studies, the original results were not replicated. Study 1 reports increased looks to a belief-congruent rather than a belief-incongruent location in a simple FB condition and random looking in a TB condition controlling

for a last-location bias. However, Study 2 (the allegedly more engaging and realistic study) fails to replicate either of these findings. Overall, both studies find more first saccades to the belief-congruent location in a TB but not in an FB condition.

Learning about failed replications can be valuable and important for a field, but in this particular case, it is not clear what we can learn from these results. There is not a single conclusion that could be unequivocally drawn from this study. The original findings were partially replicated in Study 1, but this partial replication was not observed in Study 2, making both sets of results uninterpretable. More generally, as the authors concede in the General Discussion, their failure to replicate the original results with allegedly improved methods cannot be taken as evidence that there is not such a thing as automatic Theory of Mind. If that is the case, what can we learn from this study?

More specifically, I have three main issues with this manuscript: (1) the dropout rates are extremely high for a study with adults, (2) there is no evidence that the stimuli are better or any more 'engaging' or 'realistic' than those used in previous studies, (3) trying to publishing null results when one's experimental manipulations have clearly failed is generally questionable.

1) The dropout rates in this study are abnormally high for a study with adults. Almost half the sample recruited in Study 1 and almost 2/3 in Study 2 were eliminated from analyses. The authors fail to discuss dropout rates in Study 1, and seem to make a mistake when reporting dropout rates in Study 2: on p.20 (line 43) the dropout rate is reported at 26%, but in the Participants section (p.16) it is reported at 220 out of 345 adults, which is 64% of the total sample.

In Study 2, the authors compare their dropout rates to the original studies with infants (Southgate et al. 2007) and children (Senju et al. 2010) rather than the study that was conducted with adults (Senju et al. 2009). This is very puzzling since it was this latter study on which the sample size for the study was based (reasonably, since both studies employed an adult population). More importantly, dropout rates normally differ between studies with infants and children, on the one hand, and studies with adults, on the other, so running a study with adults and having similar dropout rates to studies with infants and children strongly suggests that there is something wrong with the paradigm. Senju et al. (2009) report no neurotypical adults being excluded for failing to anticipate in their familiarization trials, and they had designed their trials to avoid this.

If the majority of adults didn't anticipate the agent's course of action in a straightforward familiarization trial, this basic result calls into question the validity of the experimental procedure. The authors themselves seem to agree with this view, saying early in the paper that "high dropout rates... make the results of existing studies difficult to interpret" (p.5). However, while the authors are ready to criticize the original studies for these methodological issues, when interpreting their own findings in the General Discussion, they do not give their high dropout rates any consideration.

2) The pitch of this replication study is that the stimuli were more 'engaging', 'realistic' and 'ecologically valid' than those used in the original studies. However, the only change in Study 1 was to use a chocolate bar instead of a ball (not entirely clear why that should be an improvement, especially with adults). The changes in Study 2 were also pretty minimal: the actor didn't wear a visor, she stood up and moved around, and left the scene than turning her head. The Krupenye paper that the authors refer to was much more convincingly 'engaging' since it had a natural looking environment where an actor was trying to hit another actor with a big stick dressed up as a monkey.

If anything, it seems like the scenes in this study were not engaging enough. The fact that most adults in both studies were unable to anticipate the agent's goal in the familiarization trials

suggests this very strongly. Given the importance of increasing participants' engagement in the task for the aim of the study, it seems clear that the authors should have piloted their stimuli and compared them with those used in the original studies. Without such a comparison (e.g., adults explicitly found these stimuli more engaging, watched them more attentively, or found them easier to follow), the conclusion to be derived in view of the present results is that the authors failed in their attempt to create better materials. It must be noted, however, that this is a failure of the present study, and not a failure to replicate previous work.

3. Related to the last point, we should consider what would have happened if this study had been an original study, and not a conceptual replication of previous work. In view of their null results, the authors would have had to admit – as most of us have had to admit often enough – that their study had failed: the manipulations they had carefully introduced to tap a certain effect simply didn't work. However, while researchers addressing new questions with new paradigms face the risk of failing, researchers only aiming to replicate previous studies seem to have found a new business model in Academia: if after a few failed replications, researchers finally manage to replicate the original results, the replicated results would be news and therefore publishable material. However, if their manipulations failed (as they clearly did here), they can always write up a new failed replication and continue to question previous findings, while not adding anything new to the literature.

I think this practice is highly questionable as it makes failed replications immune to failure (ironically) and always publishable, regardless of the possible shortcomings that would prevent publication of original studies. Since publication standards should be just as high for original work and replication studies, I cannot recommend this manuscript for publication.

Author's Response to Decision Letter for (RSOS-190068.R0)

See Appendix A.

RSOS-190068.R1 (Revision)

Review form: Reviewer 1

Is the manuscript scientifically sound in its present form?

Yes

Are the interpretations and conclusions justified by the results?

Yes

Is the language acceptable?

Yes

Is it clear how to access all supporting data?

Yes

Do you have any ethical concerns with this paper?

No

Have you any concerns about statistical analyses in this paper?

No

Recommendation?

Accept as is

Comments to the Author(s)

The changes satisfy all my points.

Review form: Reviewer 2

Is the manuscript scientifically sound in its present form?

Yes

Are the interpretations and conclusions justified by the results?

No

Is the language acceptable?

Yes

Is it clear how to access all supporting data?

Yes

Do you have any ethical concerns with this paper?

No

Have you any concerns about statistical analyses in this paper?

No

Recommendation?

Reject

Comments to the Author(s)

It wouldn't be the first time I review a paper where the authors had built a straw man of their opponent. What I had never seen before was a reviewer turned into a straw man, which is what the authors of this manuscript have tried to do with my review.

I have never said, neither in my review nor anywhere else, that failed replications have no value and should not be published. The point I tried to make in my first review (and which the authors chose to mischaracterize rather than address) is that the publication of failed replications should be subject to the same standards as original research. I agree with the authors that not publishing failed replications is problematic, but setting a lower bar for failed replications than for original studies is detrimental for the field and should also be avoided.

That the authors were applying double standards to the original AL studies they were trying to replicate and their own findings was clear in their original submission: while they criticized

earlier studies for their high dropout rates in the introduction, they failed to acknowledge their own later in the paper.

I think it is my duty as a reviewer to point out such double standards and to question the contribution of any submission, be that original research or a failed replication, if there are grounds for doing so.

In their response letter, the authors argue that they don't see any reason to assume that their study had failed – beyond failing to replicate the original results. This disagreement should be easy enough to settle: what were the aims of this study? Copied from the Introduction: 'The crucial question was whether these modified stimuli can reduce dropouts and reveal, more sensitively, automatic belief-tracking in adults.' (p.5).

The new analysis of dropout rates reported in Experiment 2 revealed that, contrary to what the authors had set out to do, their materials resulted in significantly higher dropout rates than the original study on which they were basing their paradigm. If this does not count as a failed experiment (given the goals set by the authors themselves at the start of the paper) I don't know what would, but in my book, this is a study that failed to achieve its goals and the only argument to publish it is that it's a failed replication and researchers need to learn about failed replications.

However, given that a number of failed replications have already been published using the same paradigm and questioning its reliability, what exactly does this one add to deserve publication in Royal Society Open Science? A new set of materials that results in higher (rather than lower) dropout rates than the original study? Is that something that researchers should learn about?

Such a study would never be published if it was original research, which I insist raises some very serious questions about publication standards for failed replications. Should they just get a free pass to avoid 'dangerous file-drawer problems', as the authors argue?

I am sorry but I disagree. The authors of this manuscript have failed in their own aims and other published studies have already shown that the AL paradigm is unreliable. What exactly one can learn from this study which deserves publication in a top journal is unclear to me and would be unclear to others (at least if they don't have vested theoretical interests in seeing more failed replications of ToM studies being published).

If the Editor sees enough merit in this paper to warrant publication, I would at least urge her to ask the authors to recalculate their dropout rates. This is how they explain their dropout rates in their letter:

"Please note that there are two different types of dropout rate, which seem to have led to the confusion you mention regarding different percentage values, (1) Dropouts due to technical or behavioural issue or lack of tracking behavior and (2) Dropouts due to participants failing to show anticipatory gaze in the last familiarization trials. The dropouts that are relevant in regards to the dropout analysis are those that were excluded due to the criteria originally defined by Southgate et al. (i.e. "did not look at the areas of interest (AOI) during the crucial period in the last familiarization", i.e. type 2)."

This is how they report the actual dropout numbers in the paper:

Experiment 1

"[Of a total of 217] From further analyses, 58 participants had to be excluded because they failed the inclusion criterion based on Senju et al. (2010), did not look at the areas of interest (AOI)

during the crucial period in the last familiarization (n=14) and/or test trial (n=17), or due to technical problems (n=3)."

Experiment 2

"[Of a total of 345] Forty-three participants had to be excluded from further analyses because they failed the inclusion criterion based on Senju et al. (2010), did not look at the areas of interest (AOI) during the crucial period in the last familiarization (n=118) and/or test trial (n = 50), due to technical problems (n = 3) or experimenter errors (n = 6)."

I understand that the dropout rates that need to be analyzed should not include technical problems and experimenter errors. However, those were very few compared to those that resulted from participants not engaging in the task. I therefore don't understand why the authors included much lower dropout rates in their analyses relative to what they report in the Methods sections of their experiments. Even if one was to discard dropouts due to technical and experimenter problems, their resulting dropout rates are higher than what they report in their analyses (see p.21).

Decision letter (RSOS-190068.R1)

10-May-2019

Dear Dr Kulke:

On behalf of the Editors, I am pleased to inform you that your Manuscript RSOS-190068.R1 entitled "Is implicit Theory of Mind real but hard to detect? Testing adults with different stimulus materials" has been accepted for publication in Royal Society Open Science subject to minor revision in accordance with the referee suggestions. Please find the referees' comments at the end of this email.

The reviewers and Subject Editor have recommended publication, but also suggest some minor revisions to your manuscript. Therefore, I invite you to respond to the comments and revise your manuscript.

- Ethics statement

- Data accessibility

If you wish to submit your supporting data or code to Dryad (<http://datadryad.org/>), or modify your current submission to dryad, please use the following link:
<http://datadryad.org/submit?journalID=RSOS&manu=RSOS-190068.R1>

- **Competing interests**

- **Authors' contributions**

- **Acknowledgements**

- **Funding statement**

Because the schedule for publication is very tight, it is a condition of publication that you submit the revised version of your manuscript before 19-May-2019. Please note that the revision deadline will expire at 00.00am on this date. If you do not think you will be able to meet this date please let me know immediately.

When submitting your revised manuscript, you will be able to respond to the comments made by the referees and upload a file "Response to Referees" in "Section 6 - File Upload". You can use this to document any changes you make to the original manuscript. In order to expedite the

processing of the revised manuscript, please be as specific as possible in your response to the referees.

on behalf of Dr Antonia Hamilton (Associate Editor) and Essi Viding (Subject Editor)
openscience@royalsociety.org

Associate Editor Comments to Author (Dr Antonia Hamilton):

Given the large sample sizes & pre-reg, I will not recommend rejecting this paper. however, it is important to calculate the drop-out rates correctly and to state the drop-out rates explicitly in the discussion so that it is clear to the reader that this task does have very high dropout rates and has not managed to fix this problem.

Reviewer comments to Author:

Reviewer: 1

Comments to the Author(s)

The changes satisfy all my points.

Reviewer: 2

Comments to the Author(s)

It wouldn't be the first time I review a paper where the authors had built a straw man of their opponent. What I had never seen before was a reviewer turned into a straw man, which is what the authors of this manuscript have tried to do with my review.

I have never said, neither in my review nor anywhere else, that failed replications have no value and should not be published. The point I tried to make in my first review (and which the authors chose to mischaracterize rather than address) is that the publication of failed replications should be subject to the same standards as original research. I agree with the authors that not publishing failed replications is problematic, but setting a lower bar for failed replications than for original studies is detrimental for the field and should also be avoided.

That the authors were applying double standards to the original AL studies they were trying to replicate and their own findings was clear in their original submission: while they criticized earlier studies for their high dropout rates in the introduction, they failed to acknowledge their own later in the paper.

I think it is my duty as a reviewer to point out such double standards and to question the contribution of any submission, be that original research or a failed replication, if there are grounds for doing so.

In their response letter, the authors argue that they don't see any reason to assume that their study had failed – beyond failing to replicate the original results. This disagreement should be easy enough to settle: what were the aims of this study? Copied from the Introduction: 'The crucial question was whether these modified stimuli can reduce dropouts and reveal, more sensitively, automatic belief-tracking in adults.' (p.5).

The new analysis of dropout rates reported in Experiment 2 revealed that, contrary to what the authors had set out to do, their materials resulted in significantly higher dropout rates than the original study on which they were basing their paradigm. If this does not count as a failed experiment (given the goals set by the authors themselves at the start of the paper) I don't know what would, but in my book, this is a study that failed to achieve its goals and the only argument to publish it is that it's a failed replication and researchers need to learn about failed replications.

However, given that a number of failed replications have already been published using the same paradigm and questioning its reliability, what exactly does this one add to deserve publication in Royal Society Open Science? A new set of materials that results in higher (rather than lower) dropout rates than the original study? Is that something that researchers should learn about?

Such a study would never be published if it was original research, which I insist raises some very serious questions about publication standards for failed replications. Should they just get a free pass to avoid 'dangerous file-drawer problems', as the authors argue?

I am sorry but I disagree. The authors of this manuscript have failed in their own aims and other

published studies have already shown that the AL paradigm is unreliable. What exactly one can learn from this study which deserves publication in a top journal is unclear to me and would be unclear to others (at least if they don't have vested theoretical interests in seeing more failed replications of ToM studies being published).

If the Editor sees enough merit in this paper to warrant publication, I would at least urge her to ask the authors to recalculate their dropout rates. This is how they explain their dropout rates in their letter:

"Please note that there are two different types of dropout rate, which seem to have led to the confusion you mention regarding different percentage values, (1) Dropouts due to technical or behavioural issue or lack of tracking behavior and (2) Dropouts due to participants failing to show anticipatory gaze in the last familiarization trials. The dropouts that are relevant in regards to the dropout analysis are those that were excluded due to the criteria originally defined by Southgate et al. (i.e. "did not look at the areas of interest (AOI) during the crucial period in the last familiarization", i.e. type 2)."

This is how they report the actual dropout numbers in the paper:

Experiment 1

"[Of a total of 217] From further analyses, 58 participants had to be excluded because they failed the inclusion criterion based on Senju et al. (2010), did not look at the areas of interest (AOI) during the crucial period in the last familiarization (n=14) and/or test trial (n=17), or due to technical problems (n=3)."

Experiment 2

"[Of a total of 345] Fourty-three participants had to be excluded from further analyses because they failed the inclusion criterion based on Senju et al. (2010), did not look at the areas of interest (AOI) during the crucial period in the last familiarization (n=118) and/or test trial (n = 50), due to technical problems (n = 3) or experimenter errors (n = 6)."

I understand that the dropout rates that need to be analyzed should not include technical problems and experimenter errors. However, those were very few compared to those that resulted from participants not engaging in the task. I therefore don't understand why the authors included much lower dropout rates in their analyses relative to what they report in the Methods sections of their experiments. Even if one was to discard dropouts due to technical and experimenter problems, their resulting dropout rates are higher than what they report in their analyses (see p.21).

Author's Response to Decision Letter for (RSOS-190068.R1)

See Appendix B.

Decision letter (RSOS-190068.R2)

04-Jun-2019

Dear Dr Kulke,

I am pleased to inform you that your manuscript entitled "Is implicit Theory of Mind real but hard to detect? Testing adults with different stimulus materials" is now accepted for publication in Royal Society Open Science.

on behalf of Dr Antonia Hamilton (Associate Editor) and Essi Viding (Subject Editor)
openscience@royalsociety.org

Appendix A

Dear Dr. Hamilton,

We thank you and the reviewers for your helpful feedback on the manuscript. In response, we have now carefully and thoroughly revised the paper. In particular, we have included the references and the exploratory analysis combining both datasets suggested by Reviewer 1, and have added a more detailed discussion of the three issues outlined by Reviewer 2. We now also supply clearer information on the preregistration of the current studies: Both studies were pre-registered and the preregistrations are publicly available online:

- Study 1: <https://osf.io/hj9kr/register/565fb3678c5e4a66b5582f67>
- Study 2: <https://osf.io/65pv8/register/565fb3678c5e4a66b5582f67>).

The link to the pre-registration of study 2 was previously only reported in the results section. We have now added it to the beginning of the methods section as well, where we previously only reported the link to all stimulus material (<https://osf.io/2zp46/>).

What remains, though, seems to be some deeper disagreement with Reviewer 2 regarding the informativeness and usefulness of unsuccessful replication attempts and null findings. In contrast to Reviewer 2, we firmly believe (and present corresponding arguments below) that null findings can be highly relevant, and publishing them can help researchers to know which paradigms they can build up on in future research, but also which ones they should avoid to save time and resources.

Please find a detailed list below, outlining how we addressed the reviewers' specific comments.

We hope that the manuscript is now suitable for publication in Royal Society Open Science.

Yours faithfully

Louisa Kulke, Marieke Wübker & Hannes Rakoczy

Comments to Author:

Reviewers' Comments to Author:

Reviewer: 1

Comments to the Author(s)

This paper makes an important contribution to the heated discussion about the validity of the so-called implicit tests of false belief understanding at very young ages. The contribution is indirect by testing whether adults can make sense of stimuli to which infants are subjected. The results make clear that even adults often miss the intended interpretations of the presented stimuli.

The paper neglects two recent studies using very similar stimuli for adults. Both of them provide some positive evidence for translocation studies: Low & Edwards 2018 for violation of expectation and, directly relevant, Schuwerk et al 2018 for anticipatory looking (AL). The authors need to discuss how their mostly negative results relate to these weak but positive results and what we can conclude about the reliability of these effects.

Low, J., & Edwards, K. (2018). The curious case of adults' interpretations of violation-of-expectation false belief scenarios. *Cognitive Development*, 46, 86-96.

Schuwerk, T., Priewasser, B., Sodian, B., & Perner, J. (2018). The robustness and generalizability of findings on spontaneous false belief sensitivity: a replication attempt. *Royal Society open science*, 5(5), 172273.

Thank you for pointing out these recent publications. We have added them to the manuscript (p. 5 and p. 25 & p. 26) and discuss on p. 27 that we believe that this reflects an unreliability of the tasks.

In the presentation of results a clear distinction should be made between predicted effects by different hypotheses (looking at believed location, looking at last object location, looking at head-turn location, ...) permitting use of t-test. Fortuitous findings, e.g., looking time to "box 2" for FB1 > TB1, without including FB2 in the test, need to be a-posteriori corrected. Although the overall number of participants is quite large, the numbers per condition seem insufficient to get even massive looking effects significant (e.g., the difference in first looks between FB and TB in Study 2). Perhaps one could add an additional analysis for the data of both experiments to reduce the number of relevant looking but insignificant effects.

The sample size in the current studies was computed on the basis of the original studies by Southgate et al. and Senju et al., which the current studies followed up on. Based on these effects, the power should be sufficient to detect effects. However, we agree that the original studies may have overestimated potential effects. We therefore also report Bayes factors in the Supplements (Supplement E), as Bayesian statistics allow for a comparison of the Null hypothesis and alternative hypotheses. We believe that the inconclusive findings further support the idea that the tasks are not reliable and should not be used on a single-subject level. We suggest in the discussion (p. 28) that large-scale multi-lab studies are required to reach the sample sizes to detect such small (if existent) effects.

As an analysis of the combined datasets of both studies has not been pre-registered, we refrained from conducting it so far. Due to the different AOI sizes and different stimulus materials in Studies 1 and 2, we would be cautious in conducting an analysis across both studies for saccade and raw looking time measures. However, the DLS is comparable between different tasks, as it is a relative score comparing different AOIs (see e.g. Kulke, Reiß, Krist & Rakoczy, 2018). Therefore, we conducted an exploratory analysis of the DLS across both Studies. The findings are in line with previous results. However, interestingly, the DLS in the TB1 condition is now marginally significantly positive. We have added the analysis as an exploratory analysis to p. 23 and incorporated the findings in the Discussion.

The confounding between head turn and belief-congruent looking was emphasized as an important improvement in the Introduction but the relevant data are not presented. It would be interesting to see whether this confounding does have an effect.

We have added a reference to a previous study demonstrating effects of the turn direction to the introduction (p. 9: "Previous studies suggest that the turn of the actress can significantly affect gaze patterns (Kulke, von Duhn, et al., 2018)."). As this has previously been demonstrated, we did not pre-register an analysis of the turn direction but decided to counterbalance the direction across conditions. However, we provide all data files including the variable "turn" which enables other researchers to conduct such analyses on our data.

Minor points:

On page 3 we read "These findings have revolutionized the field," which sounds odd in the context of a paper that claims that the reported findings are still open to doubt. How could unreliable data have revolutionized a field?

We have changed the wording to „drastically affected“.

The use of "clockwise" on page 9 is ambiguous. Presumably it refers to a bird's eye view of the agent's head turn.

We have changed the word and now use "left" and "right".

The text, "all participants who did make any saccade (n = 9) directed their first saccades towards the belief-congruent window, p = .004, in the TB condition," on page 14 does not match Figure 2, where all 9 looked at "box 1," which according to Figure 1 is not the belief congruent window.

Thank you for pointing this out. We have revised the figure accordingly.

Page 19: "pseudo-randomly assigned to one of the conditions (FB1, FB2, TB1, TB or FB) based on a predetermined randomised list of the conditions." Was this done once for all children or for each child individually?

A list of conditions in random order was created and the participants were assigned to each condition in the order in which they were tested.

Page 23: "In Study 2, the actor's absence from a scene during a belief induction may actually make it more difficult to track her perspective than simply turning around (see e.g., Rubio-Fernández & Geurts, 2013) However, the disappearing of the actress was in line with the study by Krupenye et al. (2016), which the stimuli were based on." Presumably the finding by Rubio-Fernandez & Geurts speaks against the use of this method in this study designed to make the tasks as easy as possible. But how does the fact that the study followed Krupenye et al alleviate this problem?

The study by Krupenye et al. found significant belief tracking effects despite the actor disappearing from the scene. We have added this information to p. 25.

The verdict, "this questions the overall suitability of anticipatory looking measures for measuring Theory of Mind," on page 24 ignores the fact that the original method (Clements & Perner 1994) of measuring AL in the traditional FB task, with text and not just based on inference from visual observations, seems to have produced highly reliable evidence of earlier understanding than answers to questions (Kulke & Rakoczy 2018). From this one might want to draw the alternative conclusion that in the non-verbal paradigms of Southgate et al children and even adults find it difficult to understand what the purpose of the agent's actions are. But once they understand this—because the story makes it clear—AL may well be a useful method.

We have added this possibility to the discussion (p. 25): "In specific, the first anticipatory looking false belief task by Clements and Perner (1994) was found to be reliably replicable (Kulke & Rakoczy, 2018a), possibly because it is easier for participants to follow (Kulke & Rakoczy, 2018b). The design of the task may therefore play a crucial role for anticipatory looking measures."

Reviewer: 2

Comments to the Author(s)

Review: "Is implicit Theory of Mind real but hard to detect? Tests with different stimulus materials"

The authors report a failed replication of anticipatory-looking false-belief studies by Southgate et al. (2007) and Surian and Geraci (2012) with adult participants. The aim of this replication study was to investigate automatic Theory of Mind using improved, more engaging materials with adults. As in other recent replication attempts of Theory of Mind studies, the original results were not replicated. Study 1 reports increased looks to a belief-congruent rather than a belief-incongruent location in a simple FB condition and random looking in a TB condition controlling for a last-location bias. However, Study 2 (the allegedly more engaging and realistic study) fails to replicate either of these findings. Overall, both studies find more first saccades to the belief-congruent location in a TB but not in an FB condition.

Learning about failed replications can be valuable and important for a field, but in this particular case, it is not clear what we can learn from these results. There is not a single conclusion that could be unequivocally drawn from this study. The original findings were partially replicated in Study 1, but this partial replication was not observed in Study 2, making both sets of results uninterpretable. More generally, as the authors concede in the General Discussion, their failure to replicate the original results with allegedly improved methods cannot be taken as evidence that there is not such a thing as automatic Theory of Mind. If that is the case, what can we learn from this study?

We believe that indeed the present findings converge with a growing body of similar findings to yield one single conclusion, namely the following: implicit AL Theory of Mind tasks are less reliable than original studies suggest. The fact that findings in the present Studies 1 and 2 diverged does not only not speak against the present approach. On the contrary, such divergent findings are exactly what you would expect if original findings were false positives and original tasks were unreliable. Again, empirically, these findings converge with other recent results from replication attempts.

What the present study with various stimulus materials adds to previous non-replications which used identical stimulus materials as the original studies is the following: it shows that the AL tasks are unreliable for different types of stimulus materials and thus that caution should be taken if other researchers want to build up their research on these unreliable tasks.

More specifically, I have three main issues with this manuscript: (1) the dropout rates are extremely high for a study with adults, (2) there is no evidence that the stimuli are better or any more 'engaging' or 'realistic' than those used in previous studies, (3) trying to publishing null results when one's experimental manipulations have clearly failed is generally questionable.

We have addressed your three main issues as outlined below.

1) The dropout rates in this study are abnormally high for a study with adults. Almost half the sample recruited in Study 1 and almost 2/3 in Study 2 were eliminated from analyses. The authors fail to discuss dropout rates in Study 1, and seem to make a mistake when reporting dropout rates in Study 2: on p.20 (line 43) the dropout rate is reported at 26%, but in the Participants section (p.16) it is reported at 220 out of 345 adults, which is 64% of the total sample.

Thank you for bringing up the issue regarding dropout rates. We agree that it is relevant to discuss these in more detail and have therefore added a more detailed discussion of the dropout rates to p. 28. We have also added a discussion of dropout rates in Study 1 to p.16. Note that Study 1 was not originally designed to investigate dropout rates (see preregistration: <https://osf.io/hj9kr/>), but Study 2, building up on the first study, set this as an aim.

Please note that there are two different types of dropout rate, which seem to have led to the confusion you mention regarding different percentage values, (1) Dropouts due to technical or behavioural issue or lack of tracking behavior and (2) Dropouts due to participants failing to show anticipatory gaze in the last familiarization trials. The dropouts that are relevant in regards to the

dropout analysis are those that were excluded due to the criteria originally defined by Southgate et al. (i.e. “did not look at the areas of interest (AOI) during the crucial period in the last familiarization”, see p. 16, bottom), i.e. type 2. This type of dropout is based on the idea that participants need to “understand” the task to be included in the analysis. Therefore, the high numbers here signify that a major number of participants does not understand these tasks. Note that this number of dropouts is comparable to other replication studies (e.g. Kulke, Johannsen, & Rakoczy, accepted; Kulke, von Duhn, et al., 2018; Schuwerk, Priewasser, Sodian, & Perner, 2018b). This further supports the idea that currently used implicit Theory of Mind tasks are unreliable.

In Study 2, the authors compare their dropout rates to the original studies with infants (Southgate et al. 2007) and children (Senju et al. 2010) rather than the study that was conducted with adults (Senju et al. 2009). This is very puzzling since it was this latter study on which the sample size for the study was based (reasonably, since both studies employed an adult population). More importantly, dropout rates normally differ between studies with infants and children, on the one hand, and studies with adults, on the other, so running a study with adults and having similar dropout rates to studies with infants and children strongly suggests that there is something wrong with the paradigm. Senju et al. (2009) report no neurotypical adults being excluded for failing to anticipate in their familiarization trials, and they had designed their trials to avoid this.

If the majority of adults didn’t anticipate the agent’s course of action in a straightforward familiarization trial, this basic result calls into question the validity of the experimental procedure. The authors themselves seem to agree with this view, saying early in the paper that “high dropout rates... make the results of existing studies difficult to interpret” (p.5). However, while the authors are ready to criticize the original studies for these methodological issues, when interpreting their own findings in the General Discussion, they do not give their high dropout rates any consideration. Thank you very much for raising some critical issues about dropout rates that we were not sufficiently clear about in the previous version. In response, we now make the following points more clearly and explicitly:

- (i) First of all, in AL tasks it has generally not been the case that inclusion/exclusion rates according to the criterion used by Southgate et al. and Senju et al. (correct anticipation in last familiarization trial) differ between children and adults (see, e.g. Kulke, von Duhn, Schneider & Rakoczy, 2018).
- (ii) We explain this more clearly now, and compare our exclusion rates to those of previous studies that used the original stimuli more generally (rather than only to Southgate et al.) on p. 26 of the manuscript.
- (iii) We completely agree that this pattern of findings “calls into question the validity of the experimental procedure”. Note that we consider the high dropout rates equally problematic for the original stimulus materials as well as the new ones developed in this study. In fact, we think that this is the main gist of the present results in concert with many other converging recent replication findings (see discussion on p. 27-28).

All in all, we thus agree that the paradigm seems unreliable. As the dropout rates are comparable to direct replications that used the original stimuli, they do not seem related to the present stimulus material. Rather, the task itself seems unreliable.

2) The pitch of this replication study is that the stimuli were more ‘engaging’, ‘realistic’ and ‘ecologically valid’ than those used in the original studies. However, the only change in Study 1 was to use a chocolate bar instead of a ball (not entirely clear why that should be an improvement, especially with adults). The changes in Study 2 were also pretty minimal: the actor didn’t wear a visor, she stood up and moved around, and left the scene than turning her head. The Krupenye paper that the authors refer to was much more convincingly ‘engaging’ since it had a natural looking environment where an actor was trying to hit another actor with a big stick dressed up as a monkey.

If anything, it seems like the scenes in this study were not engaging enough. The fact that most adults in both studies were unable to anticipate the agent's goal in the familiarization trials suggests this very strongly. Given the importance of increasing participants' engagement in the task for the aim of the study, it seems clear that the authors should have piloted their stimuli and compared them with those used in the original studies. Without such a comparison (e.g., adults explicitly found these stimuli more engaging, watched them more attentively, or found them easier to follow), the conclusion to be derived in view of the present results is that the authors failed in their attempt to create better materials. It must be noted, however, that this is a failure of the present study, and not a failure to replicate previous work.

Thank you very much for raising some crucial questions about the underlying rationale of the present stimuli. Like the stimuli in Krupenye et al., the present stimuli were created based on intuition about ecological relevance and engaging-ness. It simply seemed obvious to us, and we hope it seems equally obvious to readers, that the present stimulus material is more relevant and engaging than the (fairly sterile and boring) Southgate et al. stimuli. Of course, relevance and related notions are relative and so perhaps the present stimuli were still not relevant enough to tap into the corresponding spontaneous AL. We did already discuss this possibility in the previous version, but make this point more explicitly now (see discussion on p. 26-27).

It is interesting, by the way, that Reviewer 2 argues along the following lines: "it seems like the scenes in this study were not engaging enough. The fact that most adults in both studies were unable to anticipate the agent's goal in the familiarization trials suggests this very strongly". This is interesting because comparable levels of performance in the familiarization trials were found by Krupenye et al. (2016). In our view, this not only makes the Krupenye et al. findings difficult to interpret (because subjects who did not pass the Southgate et al. inclusion criterion were included in the main analyses), but converges with the present and many other findings in suggesting that AL tasks of the present kind, relatively irrespective of stimulus material, are not suitable for eliciting spontaneous action interpretation and thus lack validity.

3. Related to the last point, we should consider what would have happened if this study had been an original study, and not a conceptual replication of previous work. In view of their null results, the authors would have had to admit – as most of us have had to admit often enough – that their study had failed: the manipulations they had carefully introduced to tap a certain effect simply didn't work. However, while researchers addressing new questions with new paradigms face the risk of failing, researchers only aiming to replicate previous studies seem to have found a new business model in Academia: if after a few failed replications, researchers finally manage to replicate the original results, the replicated results would be news and therefore publishable material. However, if their manipulations failed (as they clearly did here), they can always write up a new failed replication and continue to question previous findings, while not adding anything new to the literature.

I think this practice is highly questionable as it makes failed replications immune to failure (ironically) and always publishable, regardless of the possible shortcomings that would prevent publication of original studies. Since publication standards should be just as high for original work and replication studies, I cannot recommend this manuscript for publication.

Regarding this issue, we respectfully and deeply disagree. We think there is some potentially dangerous ambiguity in Reviewer 2's treatment of failure. Yes, the present study –in line with a huge body of current non-replications - *fail* to replicate original finding. But this does not mean that "their study had *failed*". The latter –failure in quite a different sense- would only be true if the present study failed to investigate what it aimed to investigate. And we cannot see any reason for such an assumption.

The situation, as we see it, is this: direct replication attempts of the relevant AL tasks recently have largely yielded negative results. These results are somewhat difficult to interpret given the high

dropout rates in both original and replication studies. The present, fully pre-registered, study had two aims: first, test the generality of the null results of replication studies by using more engaging stimuli; second, reduce dropout rates with the new stimuli. Results suggest that dropout rates remain comparable to previous studies and that the original results could, again, not be replicated. This licenses an interesting conclusion: that these kind of AL tasks are not robust and valid tasks of spontaneous Theory of Mind. And it has relevant practical consequences for other researchers planning to work in the area of implicit Theory of Mind. It can help them to avoid investing time and resources (monetary and other) into projects based on unreliable and invalid tasks.

In general, we strongly disagree with Reviewer 2's views about publication of findings like the present ones. We think not only that the present findings, in light of the informative conclusions they warrant, should be published. We also think that not publishing such findings adds to dangerous file-drawer problems.

Appendix B

Dear Dr. Hamilton and Mr. Dunn,

Thank you for reviewing our submission. We are delighted to hear that the manuscript was accepted for publication in Royal Society Open Science subject to minor revisions. Please find below a report of how we have revised our manuscript in response to the editor and reviewer comments. We hope that the manuscript is now suitable for publication in Royal Society Open Science.

Yours faithfully

Louisa Kulke, Marieke Wübker & Hannes Rakoczy

Associate Editor Comments to Author (Dr Antonia Hamilton):

Given the large sample sizes & pre-reg, I will not recommend rejecting this paper. However, it is important to calculate the drop-out rates correctly and to state the drop-out rates explicitly in the discussion so that it is clear to the reader that this task does have very high dropout rates and has not managed to fix this problem.

- Response: Thank you for pointing out that the drop out rate calculations should be clarified in the analysis and the discussion section. We have clarified the dropout rates as detailed in our response to Reviewer 2, added tables displaying the number of dropouts per criterion and have expanded on this issue in the discussion, stating “In addition, it needs to be noted that dropout rates remained high despite the more engaging videos. This is in contrast to the original aim of the study to reduce dropout rates.” (p. 26), followed by an extensive discussion of potential reasons (“As dropout rates are in line with other recent implicit Theory of Mind studies, some of which were using the original stimulus by Southgate et al (Kulke, Johannsen, & Rakoczy, in press; Kulke, von Duhn, et al., 2018; Schuwerk, Priewasser, Sodian, & Perner, 2018a), the high dropout rates do not seem related to the stimulus material but rather to the task. ...”).

Reviewer comments to Author:

Reviewer: 1

Comments to the Author(s)

The changes satisfy all my points.

Reviewer: 2

Comments to the Author(s)

It wouldn't be the first time I review a paper where the authors had built a straw man of their opponent. What I had never seen before was a reviewer turned into a straw man, which is what the authors of this manuscript have tried to do with my review.

I have never said, neither in my review nor anywhere else, that failed replications have no value and should not be published. The point I tried to make in my first review (and which the authors chose to mischaracterize rather than address) is that the publication of failed replications should be subject to the same standards as original research. I agree with the authors that not publishing failed replications is problematic, but setting a lower bar for failed replications than for original studies is detrimental for the field and should also be avoided.

That the authors were applying double standards to the original AL studies they were trying to replicate and their own findings was clear in their original submission: while they criticized earlier studies for their high dropout rates in the introduction, they failed to acknowledge their own later in the paper.

I think it is my duty as a reviewer to point out such double standards and to question the contribution of any submission, be that original research or a failed replication, if there are grounds for doing so. In their response letter, the authors argue that they don't see any reason to assume that their study had failed – beyond failing to replicate the original results. This disagreement should be easy enough to settle: what were the aims of this study? Copied from the Introduction: 'The crucial question was whether these modified stimuli can reduce dropouts and reveal, more sensitively, automatic belief-tracking in adults.' (p.5).

The new analysis of dropout rates reported in Experiment 2 revealed that, contrary to what the authors had set out to do, their materials resulted in significantly higher dropout rates than the original study on which they were basing their paradigm. If this does not count as a failed experiment (given the goals set by the authors themselves at the start of the paper) I don't know what would, but in my book, this is a study that failed to achieve its goals and the only argument to publish it is that it's a failed replication and researchers need to learn about failed replications.

However, given that a number of failed replications have already been published using the same paradigm and questioning its reliability, what exactly does this one add to deserve publication in Royal Society Open Science? A new set of materials that results in higher (rather than lower) dropout rates than the original study? Is that something that researchers should learn about?

Such a study would never be published if it was original research, which I insist raises some very serious questions about publication standards for failed replications. Should they just get a free pass to avoid 'dangerous file-drawer problems', as the authors argue?

I am sorry but I disagree. The authors of this manuscript have failed in their own aims and other published studies have already shown that the AL paradigm is unreliable. What exactly one can learn from this study which deserves publication in a top journal is unclear to me and would be unclear to others (at least if they don't have vested theoretical interests in seeing more failed replications of ToM studies being published).

- Response: We apologize if the reviewer feels like a straw man. This is not what we intended. In fact, we entirely agree with the reviewer that „failed replications should be subject to the same standards as original research“. However, we disagree that a double standard is applied to the original publications and the pre-registration. In contrast, the replication tested a significantly larger number of participants, it employed two different tasks rather than just one task (as the original study did), it was pre-registered and it reports the original, as well as additional Bayesian statistics. We therefore believe that the quality of the replication is not lower than of the original study. We agree that the drop out rate is high – however, we do not believe that this is the only criterion which should be considered to judge quality.

The reviewer argues that an experiment has “failed” if it does not show initially predicted (i.e. hypothesis-conform) results. However, we believe that the replication crisis shows how important it is to report studies that do not find hypothesis-conform findings. We believe that this is not a sign of a failure of a study, but rather shows that the findings are unbiased / unaffected by the authors' previous beliefs. The Bayesian statistics used in the current manuscript can be used to compare the probability of the originally predicted and the alternative hypothesis.

The reviewer questions whether other researchers need to learn about which study manipulations lead to higher dropout rates. We believe that it can be quite relevant for other researchers to learn about high dropout rates. Firstly, it shows that the manipulation in the current study is not successful in decreasing drop out rates. This insight can prevent other researchers from spending time and money on similar adjustments that will not decrease drop outs. Secondly, taken together with other replications showing high drop out rate, the study shows that the original studies may have underestimated the dropout rate.

As we have outlined before, we think that the fact that studies with null effects were not published previously is problematic for both original and for replication research. We would therefore not apply any “double standards” to original and replication research but rather hold

the view that pre-registered studies with high sampling rates should be made available to the public, independent of whether they are original or replication research.

If the Editor sees enough merit in this paper to warrant publication, I would at least urge her to ask the authors to recalculate their dropout rates. This is how they explain their dropout rates in their letter: "Please note that there are two different types of dropout rate, which seem to have led to the confusion you mention regarding different percentage values, (1) Dropouts due to technical or behavioural issue or lack of tracking behavior and (2) Dropouts due to participants failing to show anticipatory gaze in the last familiarization trials. The dropouts that are relevant in regards to the dropout analysis are those that were excluded due to the criteria originally defined by Southgate et al. (i.e. "did not look at the areas of interest (AOI) during the crucial period in the last familiarization", i.e. type 2)."

This is how they report the actual dropout numbers in the paper:

Experiment 1

"[Of a total of 217] From further analyses, 58 participants had to be excluded because they failed the inclusion criterion based on Senju et al. (2010), did not look at the areas of interest (AOI) during the crucial period in the last familiarization (n=14) and/or test trial (n=17), or due to technical problems (n=3)."

Experiment 2

"[Of a total of 345] Forty-three participants had to be excluded from further analyses because they failed the inclusion criterion based on Senju et al. (2010), did not look at the areas of interest (AOI) during the crucial period in the last familiarization (n=118) and/or test trial (n = 50), due to technical problems (n = 3) or experimenter errors (n = 6)."

I understand that the dropout rates that need to be analyzed should not include technical problems and experimenter errors. However, those were very few compared to those that resulted from participants not engaging in the task. I therefore don't understand why the authors included much lower dropout rates in their analyses relative to what they report in the Methods sections of their experiments. Even if one was to discard dropouts due to technical and experimenter problems, their resulting dropout rates are higher than what they report in their analyses (see p.21).

- Response: We thank the reviewer for pointing out that the computation of dropouts was still not clear. We have now adjusted the methods and results section of the manuscript to clarify the drop out analysis. In order to compute the drop-out rate, we considered the number of included participants and the number of participants excluded due to the criteria used in the original study. We have specified this information in the drop out analysis section (p. 21: "In order to analyse dropout rates, only the number of participants who were excluded for not passing the original Southgate/Senju criterion of correct anticipation in the last familiarization trial and the number of participants who were included was considered."). If participants were excluded due to other criteria (technical drop outs, no looking at all, etc.), they were not included in the analysis. To make it clearer, which numbers the dropout analyses are based on, we have stated the explicit numbers in the results section (p. 21: "26 % (125 participants were included, and 43 failed the familiarisation criterion)"). We furthermore added tables to the methods sections of studies 1 and 2 that clearly display how many participants were included and excluded based on which criteria.

We agree with the reviewer that the high dropout rate should be made clear to the reader and have therefore also expanded the paragraph in the discussion to clearly state "In addition, it needs to be noted that dropout rates remained high despite the more engaging videos. This is in contrast to the original aim of the study to reduce dropout rates." (p. 26), followed by an extensive discussion of potential reasons ("As dropout rates are in line with other recent implicit Theory of Mind studies, some of which were using the original stimulus by Southgate et al (Kulke, Johannsen, & Rakoczy, in press; Kulke, von Duhn, et al., 2018; Schuwerk, Priewasser, Sodian, & Perner, 2018a), the high dropout rates do not seem related to the stimulus material but rather to the task. ...").